# A reduced ability to discriminate social from non-social touch at the circuit level may underlie social avoidance in autism

Trishala Chari[1,2], Ariana Hernandez[1], João Couto [3] &
Carlos Portera-Cailliau [1,3] ✉

Social touch is critical for communication to impart emotions and intentions. However, certain autistic individuals experience aversion to social touch. Here, we used Neuropixels probes to record neural responses to social vs. non-social interactions in somatosensory cortex, tail of striatum, and basolateral amygdala. We find that wild type mice show aversion to repeated presentations of an inanimate object but not of another mouse. Cortical neurons are modulated especially by touch context (social vs. object), while striatal neurons change their preference depending on whether mice could choose or not to interact. In contrast, *Fmr1* knockout (KO) mice, a model of autism, find social and non-social interactions equally aversive, especially at close proximity, and their cortical/striatal neurons are less able to discriminate social valence. A linear model shows that the encoding of certain avoidance/aversive behaviors in cortical neuron activity differed between genotypes. Thus, a reduced capacity to represent social stimuli at the circuit level may underlie social avoidance in autism.

The sense of touch is crucial to social communication and interaction, and animals constantly seek it, as manifested in humans by hugging, kissing, caressing, and even tickling one another. Through affiliative touch, animals offer comfort, provide inference about their internal states, and build or modify social relationships[1–5]. Brain circuits, therefore, likely evolved to prefer social touch over non-social stimuli[6,7]. On the other hand, social touch may be perceived as aversive when it is forced onto the subject[8] or when it invades our peri-personal space[9]. Moreover, certain autistic individuals actively avoid social interactions[4,10,11], perhaps because they are unable to discriminate the unique valence/salience of social stimuli, or to discriminate them from non-social stimuli.

The circuits involved in social touch are beginning to be elucidated[6,12]. In the somatosensory cortex, neural activity is modulated differently by social touch compared to non-social touch[13–16]. In rodent vibrissal somatosensory cortical (vS1), neurons

are even capable of distinguishing between mice of different sexes[17]. This suggests that vS1 is not simply responding to shape or texture but is influenced by information about context coming from other regions known to be implicated in the encoding of social affiliative touch[18–20].

Here, we wanted to address three important questions related to how behavioral responses and neural activity are uniquely modulated by social touch: First, does activity in certain brain regions reflect an animal's preference for social vs. non-social interactions? Second, do neural responses to social vs. non-social touch depend on whether the animal has a choice to interact (voluntary vs. forced touch)? This relates to the concept of personal space, the notion that touch might be more aversive when the individual has no option but to engage in it. And finally, do these circuits process social touch differently in individuals with neurodevelopmental conditions, like autism spectrum disorder (ASD), who perceive social touch as aversive?

[1]Department of Neurology, David Geffen School of Medicine at the University of California Los Angeles, Los Angeles, CA, USA. [2]Neuroscience Interdepartmental Program, David Geffen School of Medicine at the University of California Los Angeles, Los Angeles, CA, USA. [3]Department of Neurobiology, David Geffen School of Medicine at the University of California Los Angeles, Los Angeles, CA, USA. ✉e-mail: cpcailliau@mednet.ucla.edu

Using our recently developed social touch assay[21], we first identified which brain areas are recruited by social touch and then used Neuropixels silicon probes[22] to record neuronal responses to repeated interactions with either a stranger mouse or a plastic object. We focused on three relevant brain regions: vS1, because of its critical role in processing whisker-mediated touch[23] and social touch in rats[15–17]; the basolateral amygdala (BLA) because of its involvement in emotional processing and the encoding of aversive and social stimuli[24–29]; and the tail of the striatum (tSTR), which has been implicated in sensorimotor decision-making[30] and in aversion to novelty in ASD mice[31]. In addition to wild-type (WT) control mice, we examined the *Fmr1* knockout (KO) mouse model of Fragile X Syndrome, the most common single-gene cause of ASD and intellectual disability[32]. *Fmr1* KO mice manifest tactile defensiveness to repetitive whisker stimulation[33,34] and show greater avoidance/aversion to social touch than WT controls[21].

We find that WT mice display a significant aversion to object touch but not to social touch. In contrast, *Fmr1* KO mice fail to perceive the social valence by showing as much aversion to social touch as to object touch, and show unusually high aversion to social touch within their personal space. When mice can voluntarily initiate touch, neurons in vS1 and tSTR of WT mice are preferentially modulated by social touch, whereas neurons in *Fmr1* KO mice are not. In contrast, under forced touch conditions, neurons in tSTR and BLA in both WT and KO mice responded most to forced object touch, which was the most aversive for both genotypes.

## Results

### Wild-type mice show avoidance and aversive facial expressions to object touch but not to social touch

We first asked if WT mice perceive social touch differently from non-social touch. We used the same behavioral assay we recently developed[21] in which a head-fixed test mouse that can run on a polystyrene ball is exposed to repeated presentations of either an inanimate object (50 mL plastic tube) or a stranger mouse using a motorized platform (Fig. 1a). Each presentation bout lasted 10 s and consisted of a 5 s period when the platform was stopped at the touch position, and a 5 s interstimulus interval (ISI) during which the platform moved away from and back toward the test mouse (Fig. 1a, b; see "Methods"). To assess whether mice would respond differently depending on whether they could choose to engage with their visitor, presentations were either voluntary (the platform stops at a position in which the test mouse can willingly initiate contact with the visitor via its whiskers) or forced (the platform stops at a position where the snouts of the two mice are in direct physical contact). We acquired high-resolution videos and used FaceMap and DeepLabCut (DLC) to quantify changes in facial expressions (aversive whisker protraction, orbital tightening) and in running direction (see "Methods"; Fig. 1c).

We previously reported that WT mice display running avoidance and aversive facial expressions (AFEs) to forced object touch, but not nearly as much to forced social touch[21]. We once again observed that WT mice (*n* = 9) spend a significantly greater proportion of their running time showing avoidance of object touch compared to social touch (Fig. 1d; voluntary *p* = 0.012, forced *p* = 0.005). WT mice also displayed significantly more bouts of aversive whisker protraction with object touch than with social touch across all 40 presentations (Fig. 1d; voluntary: *p* = 0.049, forced: *p* = 0.021), and a higher proportion of time displaying aversive whisker protraction with object touch (Supplementary Fig. 1a, b; forced: *p* = 0.013). Thus, WT mice prefer social to non-social interactions, in line with our previous findings[21].

We next examined whether responses are different when the animal can voluntarily initiate contact compared to when they have no choice. We found that forced touch elicited more aversive whisker protraction and orbital tightening (Fig. 1d; *p* = 0.051 and *p* = 0.026,

respectively). Thus, unsolicited object touch is particularly aversive to WT mice.

### Touch context (social vs. object) can be decoded from facial expressions

Based on the above behavioral results, we hypothesized that it would be possible to accurately decode touch context from behavior videos. Hence, we trained a support vector machine (SVM) classifier on DLC labels of orofacial movements (see "Methods"; Fig. 1e), which reflect how mice engage with their environment[35–37]. We found that whiskers contributed most to decoding accuracy of touch context (social vs. object), such that the performance of the whisker-based decoder was similar to that of the all-labels decoder (Fig. 1f, g and Supplementary Fig. 1c, d; *p* > 0.05). Interestingly, during touch presentations, the all-labels DLC decoder performance rose suddenly just before the platform stopped, after whiskers first made contact (Fig. 1h and Supplementary Fig. 1e; before vs. after *p* = 0.013 for voluntary, and *p* = 0.030 for forced). Decoding accuracy for touch context was higher for forced touch compared to voluntary (Fig. 1i; *p* = 0.012). When we trained the SVM classifier only on whisker DLC labels, we also found better decoding accuracy after whisker contact (Fig. 1j, k and Supplementary Fig. 1f; *p* = 0.018 for voluntary and *p* = 0.049 for forced).

A classifier trained to discriminate touch choice (voluntary vs. forced) also showed greater decoding accuracy upon movement of the platform towards the test mouse (Fig. 1l, m and Supplementary Fig. 1g; *p* = 0.005 for social and *p* = 0.004 for object). Furthermore, we found a non-significant trend toward greater accuracy for decoding touch choice during object touch compared to social touch, which likely reflects the special aversion of WT mice to object touch when it is forced upon them within their personal space (Fig. 1n).

### Social touch engages neurons in cortical, striatal, and amygdalar circuits

Circuits involved in social behaviors are widespread throughout the brain[12,38]. We sought to survey which brain regions are uniquely modulated by social touch. We used transgenic TRAP2 mice (cFos-Cre^ERT2 × Ai14) in which the expression of tdTom is driven in an activity-dependent manner via the cFos promoter[39,40]. TRAP2 mice received repetitive presentations of either forced object (*n* = 6) or social touch (*n* = 6) for 30 min following induction with 4-hydroxytamoxifen (4-OHT) and were perfused 72 h later to quantify tdTom expression (Fig. 2a; see "Methods"). Control TRAP2 mice (*n* = 5) were induced in a no-touch condition.

Forced object and social touch induced cFos expression across many regions throughout the brain. As expected, we identified *cFos* induction in layer 2/3 of the vS1 (Fig. 2b, c). Additionally, we observed high expression of tdTom in regions associated with social behavior and/or aversion, including the nucleus accumbens (NAc), the medial amygdala (MeA) and BLA, the paraventricular nucleus of the thalamus (PVT), the periaqueductal gray (PAG), and the insular cortex (InsCx) (Supplementary Fig. 2). Importantly, tdTom expression was significantly higher after social touch than after object touch in L2/3 of vS1, the tSTR, the BLA, and the central amygdala (CeA) (Fig. 2b, c; vS1 *p* = 0.002, tSTR *p* = 0.006, BLA *p* = 0.005, CeA *p* = 0.002), as well as in anterior cingulate cortex (ACCx), InsCx, BLA, MeA, PVT and the paraventricular nucleus of the hypothalamus (PVHy) (Supplementary Fig. 2b; ACCx *p* < 0.001, NAc *p* = 0.051, InsCx *p* = 0.008, BLA *p* = 0.005, MeA *p* = 0.030, PVT *p* = 0.002). These cortical and subcortical brain regions are all known to be involved in social behavior and aversive processing[20,26,29–31,41–46]. In contrast, we did not observe significant differences between social and object touch in the density of tdTom-expressing cells in the PAG, primary motor cortex, or visual cortex (V1) (Supplementary Fig. 2b). Thus, social touch preferentially recruits neurons in a subset of brain regions.

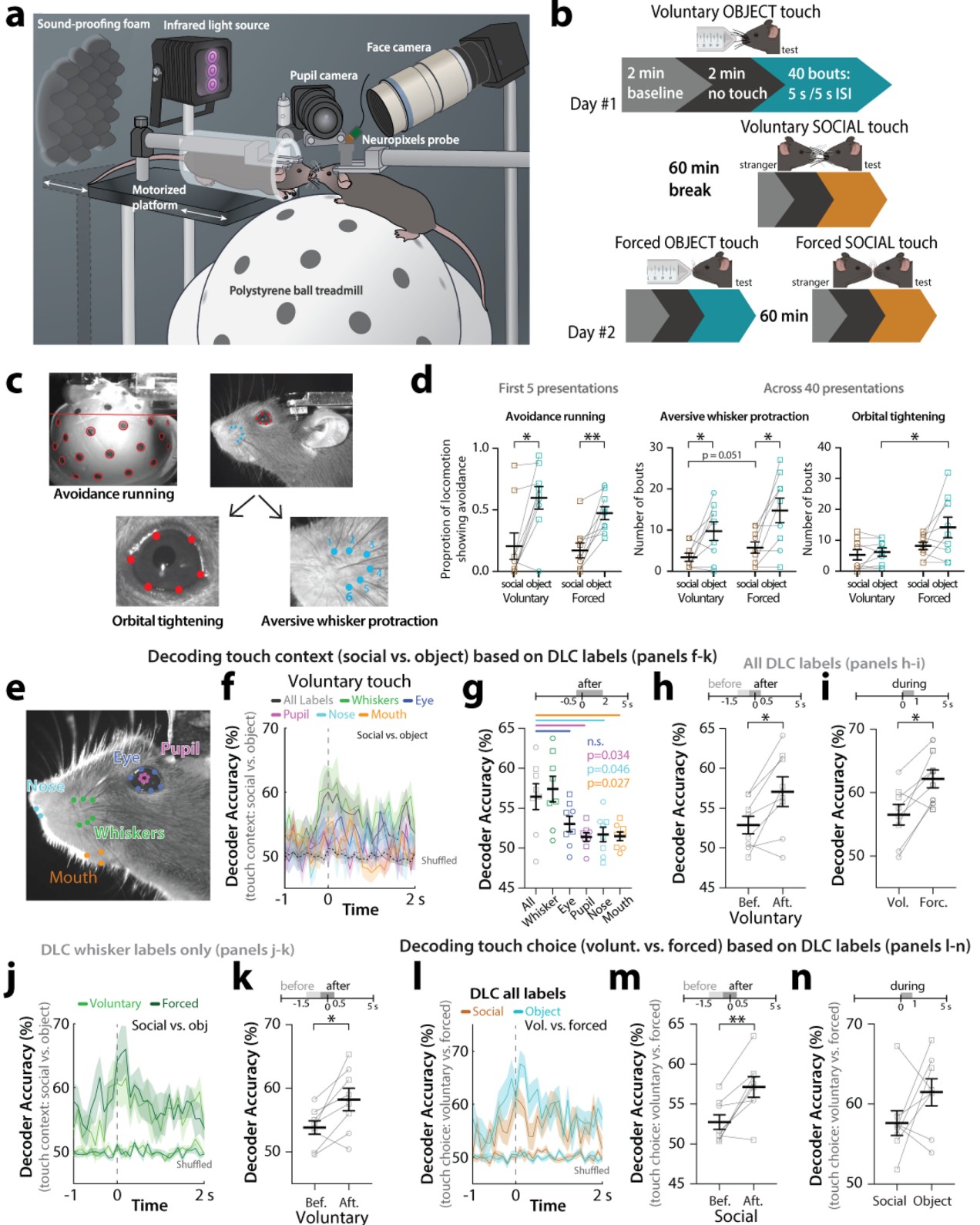

## vS1, tSTR, and BLA neurons are differentially modulated by object vs. social touch

Based on the above TRAP2 results, we chose to implant single Neuropixels probes such that their trajectory would allow us to record simultaneously from vS1, tSTR, and BLA. In this way, we could investigate how social facial touch is differentially represented from non-social touch within sensory (vS1) and emotional-related brain areas

(BLA), as well as within a sensorimotor-related brain region (tSTR). Furthermore, these brain regions have been shown to be involved during social and aversive behaviors[15,27,31,41,42,47].

We chronically implanted single-shank Neuropixels silicon probes in nine WT mice (Fig. 3a) and confirmed targeting of all 3 regions through histological reconstruction of the probe tract (Supplementary Fig. 3; see "Methods"). We used these probe trajectories and

**Fig. 1 | WT mice show avoidance and aversive facial expressions to object touch, but not social touch. a** Cartoon of behavioral assay for social facial touch. **b** Timeline of social touch behavioral assay. Mice receive 40 bouts of voluntary object and voluntary social touch, 60 min apart, on Day #1. On Day #2, mice receive 40 bouts of forced object and forced social touch. For each bout, the platform is moving for 5 s and is stopped for 5 s. Created in BioRender. Lim, K. (2025) https://BioRender.com/ppucaq3. **c** Avoidance running and different aversive facial expressions (orbital tightening and aversive whisker protraction) are quantified using FaceMap and DeepLabCut. **d** Running avoidance during the first 5 presentations of touch in WT mice (left). Number of bouts of prolonged whisker protraction (middle) and orbital tightening (right) across all 40 bouts of touch in WT mice. $N = 9$ mice; Squares = males, circles = females. $**p < 0.01$, $*p < 0.05$, two-way ANOVA with Bonferroni's. **e** The motion of DeepLabCut (DLC) labels corresponding to different facial features was used to decode touch context (social vs. object) and touch choice (voluntary vs. forced) with SVM classifiers. **f** Decoder performance of touch context (social vs. object) using DLC labels on the face in WT mice across time (from 1 s before to 2 s after the platform stops) for voluntary touch. **g** Decoder accuracy for context discrimination for using DLC labels for all

facial features or labels for individual facial features (whiskers, eye, pupil, nose, or mouth) during the time after platform movement (−0.5 s to +2 s) for voluntary touch. $p < 0.01$ for the non-parametric Kruskal–Wallis test. **h** Decoder accuracy for context discrimination using all DLC labels before (−1.5 to −0.5 s) and after (−0.5 to +0.5 s) platform movement for voluntary touch. $*p < 0.05$, parametric paired two-tailed $t$-test for (**h, i, k, m, n**). **i** Decoder accuracy for context discrimination using all DLC labels for voluntary touch and forced touch. $*p < 0.05$. **j** Decoder performance for context discrimination using only DLC whisker labels in WT mice differs across time for voluntary and forced touch (from −1 to +2 s). **k** Decoder accuracy for context discrimination using DLC whisker labels before (−1.5 to −0.5 s) and after (−0.5 to +0.5 s) platform movement for voluntary touch. $*p < 0.05$. **l** Decoder performance of touch choice (voluntary vs. forced) using all DLC labels on the mouse's face in WT mice across time for social and object touch (from −1 to +2 s). **m** Decoder accuracy for choice discrimination using all DLC labels before (−1.5 to −0.5 s) and after (−0.5 to +0.5 s) platform movement for social touch. $**p < 0.01$. **n** Decoder accuracy for choice discrimination using all DLC labels during the first second for social vs. object touch. $N = 8$ mice for (**f–n**). All data are presented as mean ± SEM.

electrophysiological landmarks to putatively assign units to vS1, tSTR, or BLA (see "Methods"; Supplementary Fig. 4a–c). Only manually curated, isolated single units were considered in the analysis (see "Methods").

We recorded the activity of single units across these three regions as mice were presented with 40 bouts of social and object touch under voluntary or forced conditions. Some neurons increased their firing in response to different presentations of touch, whereas others suppressed their firing (Supplementary Fig. 4d). We first considered the mean activity of all neurons, regardless of whether they were excited or suppressed by touch (Fig. 3b). On average, neurons across all three regions showed increased firing over baseline in response to both social and object touch, and this was apparent even before the platform stopped, because mice could initiate contact with their whiskers as the platform approached (Fig. 3c). Neurons in vS1 showed greater firing to social than object touch, whereas tSTR neurons showed stronger modulation to object touch (especially at the onset of touch) and BLA neurons did not show an obvious preference. Whether or not the animal could choose to engage in social touch also influenced neural activity in a region-dependent manner, as forced touch trials elicited a higher average response in both vS1 and tSTR than voluntary touch trials during the initial contact with the object (Fig. 3d; vS1 $p = 0.039$, tSTR $p = 0.055$).

Across the population of neurons in each region (e.g., vS1 in Fig. 3b), we observed a variety of response profiles: some neurons were suppressed by touch, some were excited transiently upon contact, others exhibited sustained firing during touch, and still others were barely modulated by touch. To compare the activity of units with such different behaviors, we performed principal component analysis (PCA) on trial-averaged activity, followed by k-means clustering of the top PCA components (see "Methods")[48–50]. This identified five significantly distinct clusters of single units in vS1, six clusters in tSTR and four clusters in BLA (Fig. 3e). Neurons that were least modulated by social/object touch (Cl. 1, Cl. 6, Cl. 12) tended to be the most abundant in all regions (Supplementary Fig. 5a). Surprisingly, neurons that were suppressed by social/object touch (Cl. 1–2, 6–7, 12–13) tended to represent a substantial proportion of the entire population (e.g., 47%, 36% and 66% in vS1, tSTR, BLA, respectively, for voluntary touch; Supplementary Fig. 5a).

Based on similarities in neural responses, we combined clusters that were strongly excited (Cl. 3–5 in vS1, Cl. 9–11 in tSTR, and Cl. 14 and 15 in BLA) or strongly suppressed (Cl. 2 in vS1, Cl. 7 in tSTR, and Cl. 13 in BLA) by touch within each region and focused on these for subsequent analyses. First, we compared differences in the modulation of their firing by touch context as an average of all 40 presentations (Fig. 3f). In vS1, we found that both excited and suppressed cells were

preferentially modulated by social touch (Fig. 3f; Supplementary Fig. 6; $p$ value range: 0.07 to <0.001). In tSTR, excited cells also showed greater modulation by social touch for voluntary presentations (Fig. 3f; $p = 0.004$), but the opposite occurred during forced interactions (Fig. 3f; Supplementary Fig. 6b; $p = 0.012$). In the BLA, only excited neurons in Cl. 14 showed significantly higher modulation by forced object touch (Supplementary Fig. 6b; $p < 0.001$). However, suppressed neurons in tSTR and BLA were not differentially modulated by social vs. object touch (Supplementary Fig. 6).

When we examined modulation by touch choice (voluntary vs. forced), we found that tSTR excited neurons showed greater modulation for forced object touch when the object was in the animal's personal space, which was the most aversive condition. Therefore, we also examined whether neural responses to forced object touch might differ during epochs when WT mice display aversion. We found that excited units in the tSTR (but not those in other brain regions, including suppressed cells in vS1) were significantly more active during bouts of avoidance/aversion compared to times when they did not display such behaviors (Fig. 3g; $p = 0.004$).

These results show that: 1. vS1 neurons can discriminate touch context and are preferentially modulated by social touch; 2. tSTR neurons care about touch choice, because they are modulated in opposite ways by social vs. object touch depending on whether mice can choose to engage; 3. tSTR and BLA neurons are preferentially modulated by forced object touch, which produces more AFEs in WT mice. Thus, for vS1 and tSTR, being able to discriminate between social vs. object touch matters for the behavioral response to stimuli that require self-initiated exploration; whereas higher tSTR and BLA firing for a stimulus predicts avoidance/aversion when it is unwanted (forced).

## Touch context can be decoded from vS1, tSTR, and BLA population activity

Considering the significant differences in neural activity between social vs. object touch, we hypothesized that SVM linear classifiers trained on Neuropixels data for all cells (see "Methods") would accurately decode touch context. Indeed, we found high decoding accuracy for both voluntary and forced conditions, even with a handful of neurons (Fig. 3h). When estimating decoding accuracy across time, accuracy was always >60% and increased sharply upon contact, particularly for forced touch, which reached >75% (Fig. 3i). We found similar results when we excluded neurons that showed large changes in baseline firing between social and object touch sessions (see "Methods"; Supplementary Fig. 7a). Decoding performance was highest for those clusters that are most strongly excited by touch, such as Cl. 5 in vS1, Cl. 10 in tSTR, and Cl. 15 in BLA, although some clusters that were

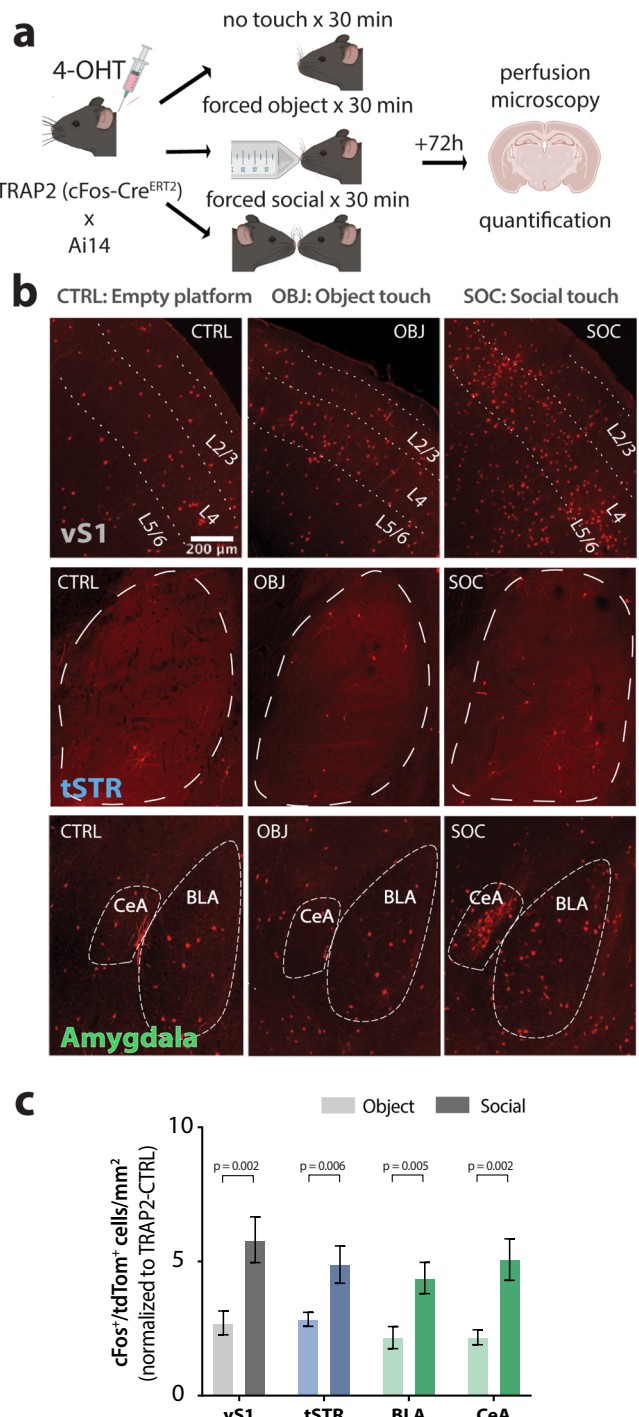

**a**

4-OHT

TRAP2 (cFos-Cre^ERT2) x Ai14

no touch x 30 min

forced object x 30 min

forced social x 30 min

+72h

perfusion microscopy

quantification

**b**

CTRL: Empty platform   OBJ: Object touch   SOC: Social touch

vS1 — CTRL / OBJ / SOC (L2/3, L4, L5/6, 200 μm)

tSTR — CTRL / OBJ / SOC

Amygdala — CTRL (CeA, BLA) / OBJ (CeA, BLA) / SOC (CeA, BLA)

**c**

cFos⁺/tdTom⁺ cells/mm² (normalized to TRAP2-CTRL)

Object / Social

vS1 p = 0.002; tSTR p = 0.006; BLA p = 0.005; CeA p = 0.002

**Fig. 2 | Differential cFos expression by forced object and social touch across vS1, tSTR, and BLA of TRAP2 mice. a** Experimental protocol for TRAP2 behavioral experiments. TRAP2 WT mice were injected with 4-OHT 30 min prior to behavioral testing, and then received either repetitive bouts of social or object touch (5 s stim, 5 s ISI) for 30 min. As a control, a subset of mice received repetitive bouts of the same duration with the platform moving but without an object or mouse present (TRAP2-CTRL). Mice were perfused 72 h later and their brains sectioned for histological analysis and quantification of cFos expression. Created in BioRender. Lim, K. (2025) https://BioRender.com/ppucaq3. **b** Example images of cFos expression in vS1, tSTR, and central amygdala (CeA) and BLA after object touch (OBJ), social touch (SOC), or no touch (empty platform; CTRL) (scale bar = 200 μm). **c** Density of cFos-expressing (tdTom⁺) cells per mm² for each brain region, normalized against CTRL density. *$p < 0.05$, normality was tested with the D'Agostino and Pearson test, followed by unpaired two-tailed non-parametric Mann–Whitney or parametric $t$-test for each brain region. For object touch, $n = 35$ vS1, $n = 28$ tSTR, $n = 30$ BLA, and $n = 25$ CeA sections. For social touch, $n = 36$ vS1, $n = 24$ tSTR, $n = 39$ BLA, and $n = 29$ CeA sections. Data presented as mean ± SEM.

## Forced touch recruits the highest proportion of object-preferring neurons in the BLA

The fact that neural activity in vS1, tSTR, and BLA showed differential modulation by social vs. object touch raises the possibility that certain neurons may exhibit a true preference for either object or social touch. We used receiver operating characteristic (ROC) analysis to categorize object-preferring and social-preferring cells (Fig. 4a–c and Supplementary Fig. 8a, b; see "Methods"). We found that at least 10% of units that were excited by touch showed a significant preference for social touch (greater than expected by chance; see "Methods"), irrespective of the brain region (Fig. 4d). In vS1, there was a similar proportion of object- and social-preferring units regardless of whether interactions were voluntary or forced (17-23%; Fig. 4d, e). In contrast, we found a higher proportion of object-preferring cells during forced interactions in the tSTR and in the BLA (Fig. 4e; $p = 0.055$ and $p = 0.003$, respectively). Together with the greater modulation of excited cells in tSTR and BLA by forced object touch, which triggers the most aversion in test mice, these findings further support the notion that these two regions are implicated in aversion to touch within the animal's personal space (when it has no choice but to engage).

## *Fmr1* KO mice do not discriminate social valence and find social interactions in their personal space more aversive than WT controls

To further understand the relationship between neural activity and behavioral responses to social vs. object touch, we next turned to *Fmr1* KO mice, because we recently discovered that they show similar aversion to social and object touch[21]. Thus, we hypothesized that neural activity may not be differentially modulated by touch context in this model of autism. We once again found that *Fmr1* KO mice ($n = 10$ mice) manifest similar levels of avoidance running and AFEs to social and object touch (Fig. 5a, b; $p > 0.05$ for all). When assessing how mice responded depending on whether they could choose to engage or not, we found that *Fmr1* KO mice displayed more aversive whisker protraction to object touch during forced presentations compared to voluntary interactions (Fig. 5b; $p = 0.009$; Supplementary Fig. 9a; $p = 0.012$).

When comparing across genotypes, we found *Fmr1* KO mice showed significantly more AFEs (whisker protraction and orbital tightening) to social touch than WT controls (Fig. 5c; $p = 0.003$). Interestingly, AFEs were significantly greater in *Fmr1* KO mice for forced social touch than for voluntary social touch (Fig. 5c; $p = 0.003$). Thus, social touch seems especially bothersome to *Fmr1* KO mice when it takes place within their personal space.

strongly suppressed by touch also showed decoding accuracies well above chance, like Cl. 7 in tSTR (Supplementary Fig. 7b, c).

We previously found that social and object encounters led to different levels of running avoidance in WT and *Fmr1* KO mice[21], and because locomotion impacts neural responses dramatically[51,52], we wondered whether locomotion is responsible for our ability to decode touch context with high accuracy. We trained a social vs. object decoder on both locomotion and neural activity and observed that the decoding accuracy was similar to that of a neural-only decoder (Supplementary Fig. 7d, magenta and black curves, respectively). Critically, decoding accuracy was brought to chance levels when neural activity, but not locomotion, was shuffled in time (Supplementary Fig. 7d, blue curve). We conclude that locomotion does not contribute to our ability to decode social from object touch from neural activity.

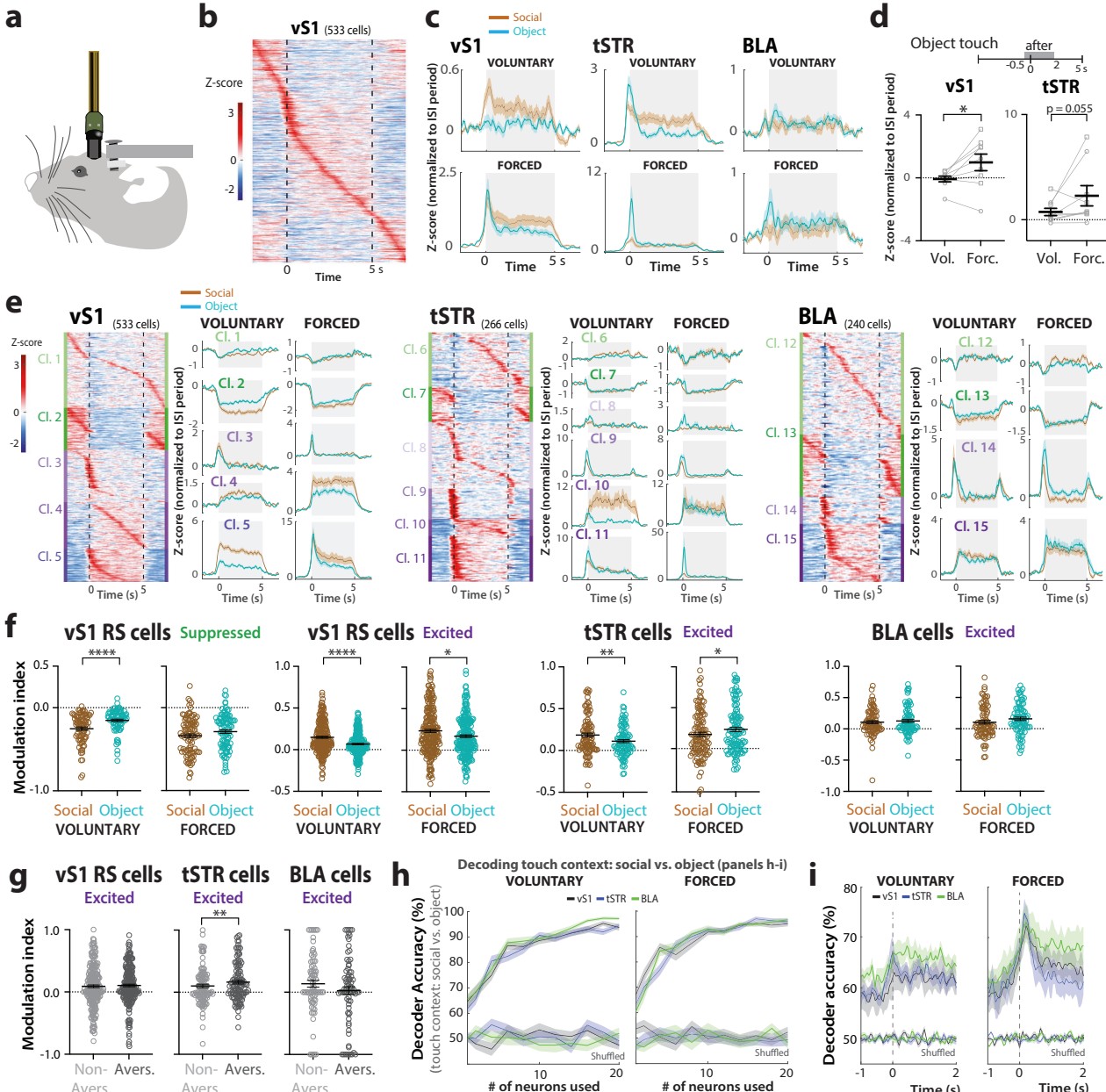

**Fig. 3 | Neurons in vS1, tSTR, and BLA show differential modulation by social vs. object touch. a** Cartoon of a mouse chronically implanted with a Neuropixels probe (not to scale). **b** Example heatmap of all vS1 cells (n = 533) sorted by peak of trial-averaged, z-scored PSTHs. The scale bar denotes z-score values. **c** Trial-averaged z-scores normalized to period before touch (ISI) for all vS1 regular-spiking (RS), tSTR, and BLA cells from nine WT mice during social (brown) and object touch (blue) for voluntary (top) and forced conditions (bottom) across time (−2 to +7 s). **d** Mean z-scores during contact (−0.5 to +2 s) for voluntary/forced object touch across different mice. *p < 0.05 for paired parametric two-tailed t-test (n = 9 mice). **e** Left: Heatmap of trial-averaged PSTHs for voluntary touch (mean of all touch presentations) for all cells split by clusters (PCA/k-means) and sorted by peak firing in time within each cluster. Right: Z-scores for all neurons of individual clusters, sorted by suppressed (green) to excited (purple). Time 0 s denotes when the platform stops. **f** Modulation index of vS1 RS excited (n = 227/220) and suppressed

cells (n = 79/98), tSTR excited cells (n = 99/116), and BLA excited cells (n = 81/79). ****p < 0.0001, **p < 0.01, *p < 0.05 for paired parametric two-tailed t-test. Symbols represent single cells (n = 9 mice). **g** Modulation index of excited cells in vS1 (n = 220), tSTR (n = 116), and BLA (n = 79) during forced object presentations when WT mice show aversion (Avers.) or not (Non-Avers.). **p < 0.01 for paired parametric two-tailed t-test. **h** Decoder accuracy for touch context (social vs. object) based on activity of cells in each region during voluntary and forced touch (0–5 s). Up to 20 neurons were randomly chosen from each brain region. As a control, we tested decoder accuracy when the context identity was shuffled in 80% of presentations. **i** Decoder accuracy for touch context based on activity of 10 randomly selected cells in each region per animal across 50 ms bins throughout voluntary and forced touch (−1 to +2 s). Data presented as mean ± SEM for (**d, f, g**). Line and shaded area correspond to mean ± SEM for (**e, h, i**).

## Orofacial movements from *Fmr1* KO mice are not as good at decoding touch context as those of WT mice

Considering how *Fmr1* KO mice show similar behavioral responses to social and object touch (Fig. 5a, b), we hypothesized that SVM classifiers trained on their facial expressions would show lower accuracy in discriminating touch context (social vs. object) compared to WT mice.

Indeed, we found that decoder performance from all DLC labels did not change in *Fmr1* KO mice following the movement of the platform towards the test mouse for voluntary touch, unlike what we had observed for WT mice (Fig. 5d, e; KO p > 0.05, WT vs. *Fmr1* KO p = 0.043). However, under forced touch conditions, the decoder

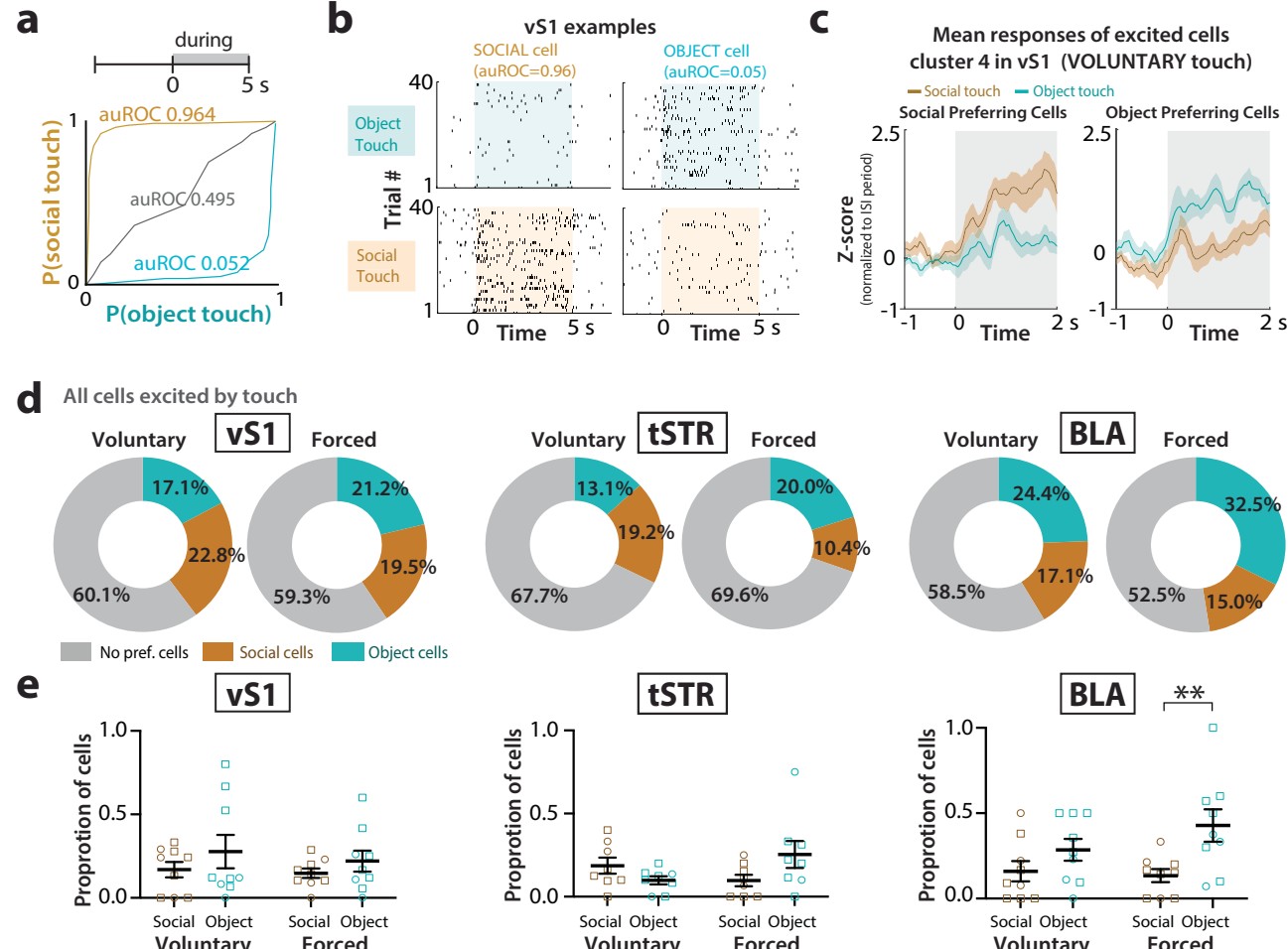

**Fig. 4 | Higher proportion of object-preferring cells in tSTR and BLA. a** Example ROC curves (and corresponding ROC values) for a significant social-preferring cell, an object-preferring cell, and a neutral cell in vS1 (from cluster 4). **b** Spike rasters across each presentation of social touch and object touch for the same example social-preferring and object-preferring cells in vS1 shown in (**a**). **c** Mean z-score firing rates of the same example social-preferring and object-preferring cells in vS1 during voluntary object touch and social touch. The line and shaded area correspond to mean ± SEM. **d** Percentage of object-preferring and social-preferring cells in vS1, tSTR, and BLA during voluntary and forced touch as total of all cells. **e** Proportion of object-preferring and social-preferring cells in vS1, tSTR, and BLA for individual mice ($n = 9$ WT mice). **$p < 0.01$ for two-way ANOVA with Bonferroni's. Squares = males, circles = females. One mouse was excluded according to ROUT's analysis for tSTR. Data presented as mean ± SEM.

performance increased similarly for WT and *Fmr1* KO mice after touch (*Fmr1* KO before vs. after $p = 0.042$; Supplementary Fig. 9b, c).

We then wondered whether classifiers trained on aversive behaviors might also perform differently in both genotypes. We combined behavior data for running avoidance, AFEs, and eye saccades (associated with avoidance in ASD[53,54]), and the resulting decoder showed better performance for WT mice under voluntary touch conditions (Fig. 5f, g; $p = 0.046$), though not for forced conditions (Supplementary Fig. 9d, $p > 0.05$).

We also tested the performance of classifiers trained on orofacial movements from *Fmr1* KO mice at discriminating touch choice (voluntary vs. forced). Just as for WT mice, we observed a significant increase in decoder accuracy after initial contact in *Fmr1* KO mice ($p = 0.032$; Supplementary Fig. 9e, f), and just as with WT mice decoding touch context was significantly higher for object touch than for social touch in *Fmr1* KO mice ($p = 0.002$; Supplementary Fig. 9g), presumably because of their higher aversion to forced object touch (Fig. 5b).

### vS1, tSTR, and BLA neurons in *Fmr1* KO mice do not distinguish between voluntary social vs. object touch

We next used Neuropixels to record from units in vS1, tSTR, and BLA of *Fmr1* KO mice, and determine the individual clusters (Supplementary Fig. 10a). The proportion of units in various clusters differed between

WT and *Fmr1* KO mice (Supplementary Fig. 5). For example, in vS1 there were significantly fewer cells in Cl. 2 and more in Cl. 5 in *Fmr1* KO mice, which are the clusters that, in WT mice, are the most modulated by social touch (Supplementary Fig. 5b). With regards to touch context, neurons in *Fmr1* KO mice were similarly modulated by social and non-social touch under voluntary conditions, unlike WT mice where neurons in vS1 and in tSTR were significantly more modulated by social touch (Supplementary Fig. 10b, c). As such, when comparing the difference in modulation (*Δmodulation*) of excited neurons between social and object touch, we found that *Fmr1* KO mice uniformly had significantly smaller magnitudes across brain regions than WT mice (Fig. 6a; voluntary: vS1 excited $p = 0.013$, tSTR excited $p = 0.038$), which matches the behavioral responses under voluntary conditions. However, under forced touch conditions, neurons excited by touch in the tSTR and BLA of *Fmr1* KO mice were more strongly modulated by object touch than by social touch (Fig. 6a; Supplementary Fig. 10b, c), similar to what we had seen in WT mice (Fig. 3f).

When comparing the mean firing rate for neurons in these brain regions across the two genotypes, we only found significant differences for suppressed neurons in vS1 responding to voluntary social touch (they fired even less in WT mice; Supplementary Fig. 11a) and for excited cells in vS1 responding to forced object touch (they fired more

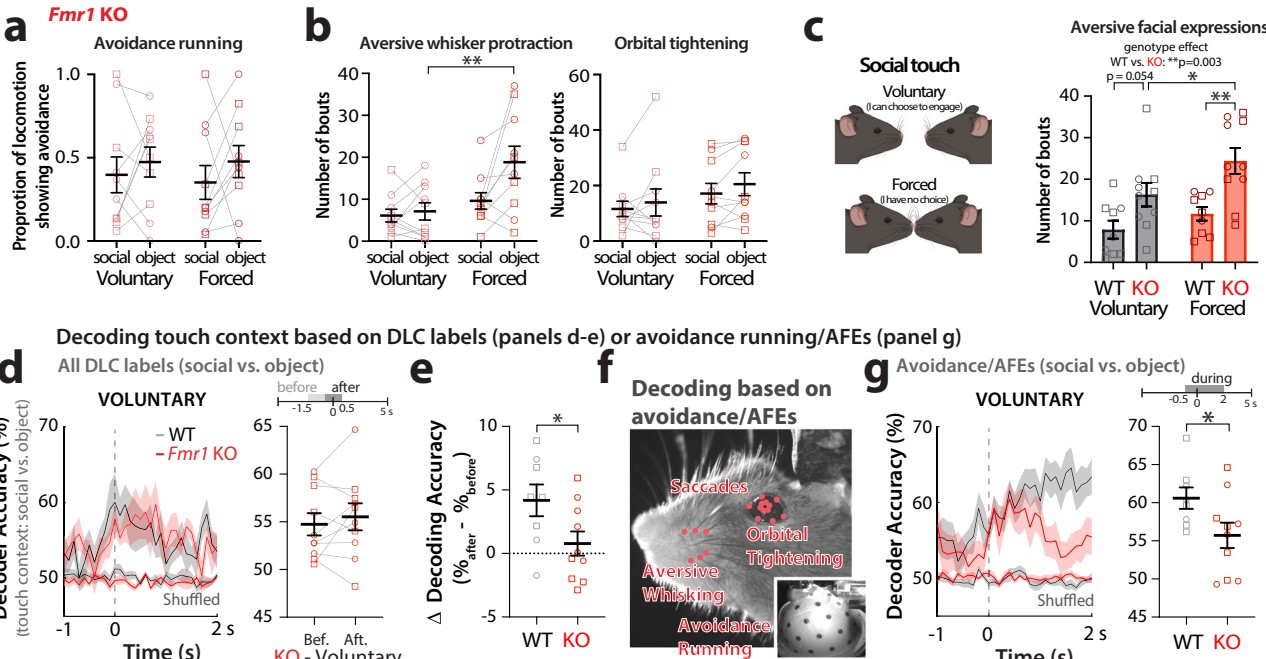

**Fig. 5 | *Fmr1* KO mice show similar aversion to social and object touch and find social interactions within their personal space, particularly aversive. a** Running avoidance during the first 5 presentations of touch in *Fmr1* KO mice (*n* = 10) for social touch and object touch. **b** Number of bouts of prolonged whisker protraction (left) and orbital tightening (right) in *Fmr1* KO mice across all 40 presentations of social or object touch in *Fmr1* KO mice. Squares = males, circles = females. \*\**p* < 0.01 for two-way ANOVA with Bonferroni's. **c** Left: Cartoon of how test mice might perceive touch choice (voluntary vs. forced touch). Created in BioRender. Lim, K. (2025) https://BioRender.com/ppucaq3. Right: Number of bouts of AFEs (aversive whisker protraction and orbital tightening) for *Fmr1* KO and WT mice during voluntary vs. forced social touch. \*\**p* < 0.01 and \**p* < 0.05 for two-way ANOVA with Bonferroni's. **d** Left: Decoder performance for touch context (social vs. object) using all DLC labels on the mouse's face in *Fmr1* KO and WT mice across time (−1 to

+2 s after platform stops). Right: Decoder accuracy for touch context before (−1.5 to −0.5 s) and after (−0.5 to +0.5 s) in *Fmr1* KO mice. Squares = males, circles = females. \**p* < 0.01 for unpaired parametric two-tailed *t*-test. **e** Change in decoder accuracy (% after minus % before) for touch context using all DLC whisker labels in *Fmr1* KO and WT. Squares = males, circles = females. \**p* < 0.05 for unpaired parametric *t*-test. **f** Avoidance running, saccades, and AFEs were used to decode touch context with SVM classifiers. **g** Left: Decoder accuracy for touch context based on aversive behaviors in WT and *Fmr1* KO mice (from −1 s to +2 s after platform stops). Right: Mean decoder accuracy (−0.5 to +2 s) for individual mice. Squares = males, circles = females. \**p* < 0.01 for unpaired parametric two-tailed *t*-test. Data presented as mean ± SEM. *N* = 9 WT and 10 *Fmr1* KO mice for (**a**–**c**). *N* = 8 WT and 10 *Fmr1* KO mice for (**d**, **e**, **g**).

in KO mice; Supplementary Fig. 11b). Interestingly, excited neurons in all 3 regions in *Fmr1* KO mice tended to fire more for object touch under forced touch conditions, whereas there was no difference in firing for WT mice between social and object touch (Supplementary Fig. 11b).

We also found that twice as many clusters showed significant differences in modulation index between social and object touch in WT mice (Cl. 2, 4, 5, 10 for voluntary interactions and Cl. 4, 9, 11, 14 for forced interactions; Supplementary Fig. 6) than in *Fmr1* KO mice (Cl. 3, 5 for voluntary interactions and Cl. 10, 11 for forced interactions; Supplementary Fig. 12). When we compared the change in modulation index between social and object touch across genotypes, we found that neurons in WT mice tended to have greater modulation by social touch than those in *Fmr1* KO mice (the difference was significant for Cl. 3, Cl. 5, and Cl. 10 for voluntary touch, and for Cl. 4, 5 for forced touch; Supplementary Fig. 13). In fact, many clusters across different brain regions in *Fmr1* KO mice tended to show the opposite preference compared to WT mice (i.e., a preference for object touch), especially under forced touch conditions, as seen for neurons in Cl. 1, Cl. 5, Cl. 9, Cl. 10, and Cl. 12.

In line with these results, SVM classifiers trained on neural data from *Fmr1* KO mice showed reduced decoding accuracy of touch context compared to WT mice (Fig. 6b, c; voluntary touch: tSTR *p* = 0.114 and BLA *p* = 0.001; similar results for forced touch; Supplementary Fig. 14a). We considered that differences in locomotion between WT and *Fmr1* KO mice could partially explain these results; however, when we repeated this analysis but included only mice that

were consistently running throughout all 40 presentations (*n* = 4–5 WT and *n* = 5 *Fmr1* KO), we found that decoder accuracy was still higher in WT mice in all three regions (Supplementary Fig. 14b). Moreover, the fact that locomotion does not contribute to decoding accuracy (Supplementary Fig. 7d) suggests that potential differences in locomotion between WT and *Fmr1* KO mice are not likely to explain the differences in decoding accuracy between genotypes.

Overall, these results mirror our behavioral observations and argue that circuits from *Fmr1* KO mice fail to discriminate between social and object touch under voluntary conditions; however, when choice is removed (forced touch), *Fmr1* KO mice demonstrate greater modulation of neurons in BLA and tSTR by object touch, which is the most aversive.

### Neurons in vS1 of WT mice, but not *Fmr1* KO mice, show a preference for social touch

We next used ROC analyses to calculate the proportion of social-preferring and object-preferring excited neurons in vS1, tSTR, and the BLA of *Fmr1* KO mice (Fig. 6d, Supplementary Fig. 15). When comparing the cumulative distributions of ROC values (auROC) for all excited neurons in vS1, we found a significantly lower social preference in *Fmr1* KO mice compared to WT controls (Fig. 6d; voluntary *p* = 0.002, forced *p* = 0.045). Only the BLA showed a significantly higher proportion of object-preferring cells (Supplementary Fig. 15b; *p* = 0.010), just as we had seen in WT mice (Fig. 4d). Thus, vS1 of *Fmr1* KO mice seems generally less able to differentiate between social and non-social stimuli.

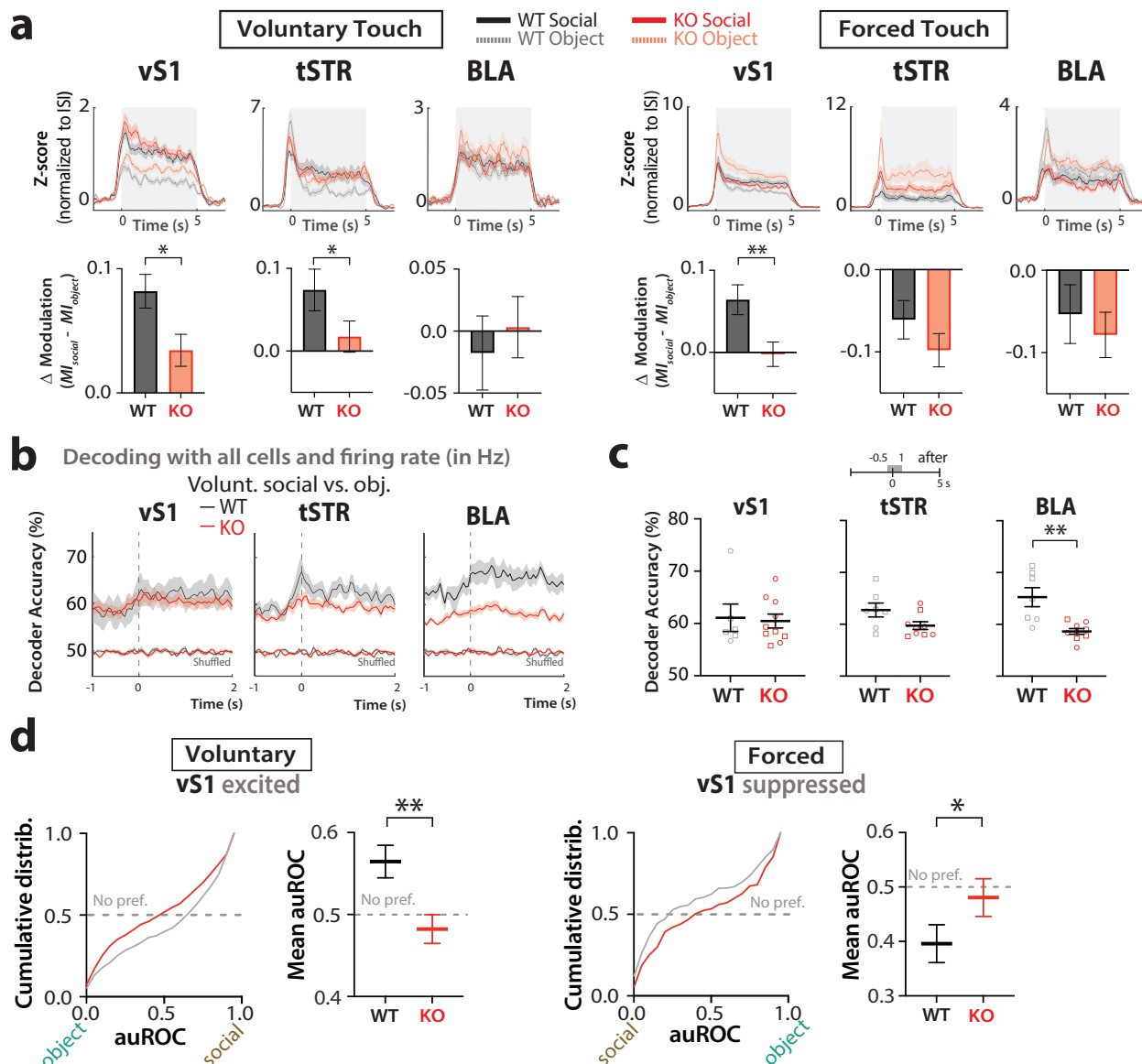

**Fig. 6 | Lack of modulation of neural activity by touch context in *Fmr1* KO mice during voluntary presentations and reduced social preference compared to WT mice. a** Top: *Z*-score activity (normalized to ISI period) of excited cells in vS1 (RS only), tSTR, and BLA during social touch and object touch, for voluntary (left) and forced presentations (right). Bottom: Corresponding Δmodulation ($MI_{social}-MI_{object}$) by social vs. object touch of cells in vS1, tSTR, and BLA of WT and *Fmr1* KO mice *$p < 0.05$, **$p < 0.01$ for unpaired two-tailed non-parametric Mann–Whitney or parametric *t*-test. For WT excited cells, vS1 $n = 227/220$, tSTR $n = 99/116$, and BLA $n = 81/79$. For *Fmr1* KO excited cells, vS1 $n = 307/331$, tSTR $n = 178/202$, and BLA $n = 105/128$. **b** Decoder accuracy for touch context, averaged across mice, based on activity of 10 randomly selected cells in vS1, tSTR, and BLA (50 ms time bins) during the presentation period of voluntary touch (−1 to +2 s).

**c** Mean decoder accuracy for touch context based on activity of 10 randomly selected cells in vS1, tSTR, and BLA (−0.5 s to +1 s after that platform stops) for individual WT and *Fmr1* KO mice. Squares = males, circles = females. **$p < 0.01$ for unpaired parametric two-tailed *t*-test. **d** Cumulative distribution (left) and mean auROC values (right) for excited and suppressed cells in vS1 (voluntary and forced touch, respectively) in WT and *Fmr1* KO mice. For excited cells, auROC values above 0.5 correspond to social preference, while values below 0.5 correspond to an object preference; the opposite is true for suppressed cells; auROC values of 0.5 correspond to no preference. **$p < 0.01$, *$p < 0.05$ for unpaired parametric two-tailed *t*-test. $N = 227$ cells for WT voluntary vS1 excited, 98 for WT forced vS1 suppressed, 307 for *Fmr1* KO voluntary vS1 excited, and 107 for *Fmr1* KO forced vS1 suppressed. Data presented as mean ± SEM.

## Neural activity in vS1 relates to avoidance behaviors and AFEs

To determine the extent to which cortical, striatal, and amygdalar population dynamics integrate and encode behavioral responses to social or object touch, we implemented a linear encoding model (see "Methods"). The model's matrix included both discrete events (e.g., social or object touch, running or stationary, aversive whisker protraction present or absent) and continuous behavior variables (e.g., orbital area, whisker movements, pupil size). In total, we used 13 such regressors (see "Methods"). We focused on forced touch because it is the presentation type that elicited the most aversive responses. We

then calculated normalized $\beta$ weights (normalized to the sum of all weights for each cell) corresponding to individual regressors (aversive and other behaviors, as well as stimulus context) that provide an estimate of how well each regressor predicts the mean firing rate of individual neurons on each trial and for the three brain regions (Fig. 7a). The model identified neurons whose activity nicely matched either stimulus context (social vs. object), running, or individual AFEs (see examples in Fig. 7b, top). Critically, individual neurons had stronger weights for certain specific regressors but not others, signifying that they were preferentially modulated by those regressors

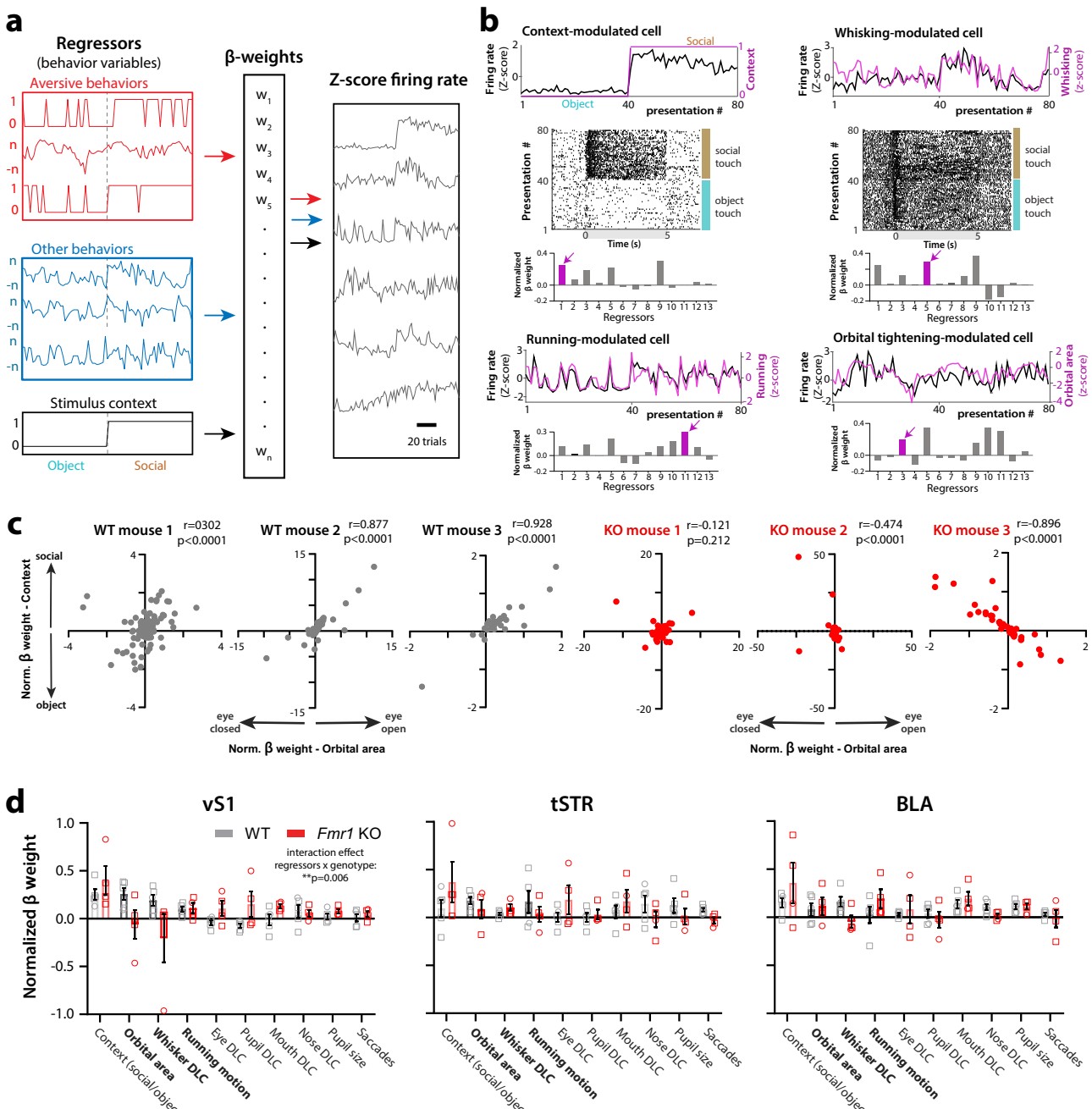

**Fig. 7 | Activity in vS1 encodes touch context and several aversive behaviors. a** A linear encoding model was built using both discrete and continuous variables (regressors) that included both aversive and other (non-aversive) behaviors, as well as the stimulus context (social or object). This generated $\beta$ weights for each regressor that determined how well each behavior variable was encoded by the activity of a particular neuron (expressed as a Z-score of firing rates) on a trial-by-trial basis. **b** Four example neurons that were uniquely modulated by touch context, whisking, running, and orbital tightening, respectively. For each cell, we show an overlay of the mean firing rate of each neuron (z-score) with the changes in the corresponding regressor (top) and the normalized beta weights of each cell for all regressors (bottom). Regressors: 1. Context (social/object), 2. Running avoidance (binary), 3. Orbital area, 4. Whisker protraction (binary), 5. Whisker DLC, 6. Eye DLC, 7. Pupil DLC, 8. Mouth DLC, 9. Nose DLC, 10. Pupil DLC, 11. Running motion, 12.

Saccades; 13. Locomotion (binary). The regressor in magenta is the one corresponding to the trace above. A raster of firing for all 80 presentations is also shown for the top two neurons. **c** Relationship between normalized $\beta$ weights for context and orbital tightening for three example WT mice and three *Fmr1* KO mice. Each symbol represents a different neuron. **d** Mean normalized $\beta$ weights across 10 different regressors (binary regressors excluded, except "context") for neurons in vS1, tSTR, and BLA for WT and *Fmr1* KO mice ($n = 5$ and 4 mice, respectively). There is a significant interaction effect (regressors × genotype) in vS1 ($p = 0.006$; ANOVA). The three regressors corresponding to avoidance running and the two main AFEs that we tracked behaviorally are shown in bold ($p = 0.064$ for orbital tightening, $p = 0.111$ for whisker DLC; two-tailed Mann–Whitney test). Data presented as mean ± SEM.

(Fig. 7b, bottom). Moreover, for individual WT mice, we found significant positive correlations between the $\beta$ weight of single neurons for social context and their $\beta$ weight for orbital area, which relates to one of the AFEs, orbital tightening (see three examples in Fig. 7c).

Hence, neurons whose activity strongly encoded object touch tended to also encode orbital tightening (smaller eye area). Interestingly, in *Fmr1* KO mice, the vS1 population showed the inverse correlation (Fig. 7c), meaning that neurons with high $\beta$ weight for social context

had low $\beta$ weight for orbital area. One caveat is that, because we use Ridge regression for model fitting (see "Methods"), we cannot distinguish contributions from individual regressors (see "Discussion").

When we compared normalized $\beta$ weights for different regressors in vS1 of WT mice, we found higher values for those related to context and aversive behaviors (e.g., orbital area, whisker movement) than for other non-aversive behavior variables like mouth and nose movements or pupil size (Fig. 7d). In the tSTR, $\beta$ weights were also higher for AFEs and running motion than for other regressors. In the BLA, there was more variability, either because of the lower number of recorded neurons compared to vS1 or because neurons in the BLA are less selective for these behaviors (consistent with their reduced modulation by social touch). When we compared the normalized $\beta$ weights for different regressors, we found genotype differences (some values were higher for WT and others higher for KO mice), and there was a significant interaction effect (regressors × genotype) in vS1 ($p = 0.006$; ANOVA), but not in tSTR or BLA. Additionally, we found higher values in WT mice than in *Fmr1* KO mice for the two main AFE regressors, orbital area and whisker DLC, although this did not reach statistical significance ($p = 0.064$ and $0.111$, respectively; non-parametric Mann–Whitney test; $n = 5$ WT and 4 *Fmr1* KO mice; Fig. 7d). We conclude that neurons in vS1 of WT mice not only encode touch context but also have strong weights for certain AFEs. In agreement with our decoding analysis (Fig. 6), neurons in vS1 of *Fmr1* KO mice seem to have altered coding for AFEs and show weaker or negative weights for those variables.

## Discussion

The main goal of this study was to investigate how different brain circuits are modulated by social facial touch and how this relates to aversive responses. We focused on both the context of touch (the social valence of touch) and the importance of having a choice to engage in social touch (the notion of peri-personal space).

Our main findings can be summarized as follows (Supplementary Fig. 16): 1. WT mice tolerate social touch but show aversion to object touch, especially at close range (forced touch). 2. vS1 neurons care about touch context and are preferentially modulated by social touch. 3. In contrast, tSTR and BLA neurons care most about touch choice, firing preferentially to object touch when it occurs in the animal's personal space. Because forced object touch elicited more avoidance/AFEs, we surmise that activity in BLA and tSTR relates to behavioral responses when the animal is forced to interact; 4. *Fmr1* KO mice show similar degrees of avoidance/AFEs to both social and object touch, which means they are unable to distinguish, at the behavioral level, the social valence of an interaction; 5. Consistent with this, neural activity in all three regions in *Fmr1* KO mice was less able to discriminate between social vs. object touch. Moreover, *Fmr1* KO mice showed striking aversion to forced touch (particularly social touch), and activity in the BLA tracked this, suggesting that maladaptive responses to stimuli in the animal's peri-personal space map onto amygdalar circuits; 6. Findings of the linear encoding model support the conclusion that neuronal activity in WT mice encodes not only touch context but also certain AFEs, whereas this relationship is weaker in *Fmr1* KO mice.

Our cFos expression data suggest that processing of social touch involves circuits distributed across many brain regions. We have identified several interesting brain regions worth exploring in the context of social touch. Notably, regions such as the InsCx, PAG, or the PVHy had already been implicated in social behaviors[55–58]. Future studies should assess whether these other brain regions can also distinguish social from non-social tactile inputs, or whether this is dependent on an animal's ability to voluntarily engage in touch.

The fact vS1 neurons still showed a social preference when the animal had no choice but to engage, suggests that the valence of the social stimulus matters to cortical neurons. Thus, vS1 neurons are not solely responding to a difference in shape/texture between a plastic tube and a mouse, but also care about valence, presumably relying on inputs from other brain regions that encode pleasurable aspects of social touch[18–20]. Furthermore, the neural activity in vS1, but not the activity of tSTR and BLA, aligns well with how WT mice tolerate social touch. Without this neural discriminability, we surmise that animals would no longer differentiate social from non-social stimuli behaviorally, as occurs in *Fmr1* KO mice (see below). Still, it is possible that some of the differences in neural activity we observed between presentations of a plastic tube and a live mouse may not just reflect the social vs. non-social context, but also be related to differences in texture, smell, or movement of the visitor mouse.

Our ROC analysis identified a significant proportion of neurons in all three regions that exhibited a clear preference for either social or object touch. Presently, it is unclear whether these populations are always non-overlapping or whether some population drift could exist, such that a cell that exhibits a clear social preference, for example, might then show a preference for object touch at a later time. We speculate that such drift does exist, at least in vS1, depending on the inputs that a neuron receives.

Our linear encoding model confirmed that AFEs are strongly encoded in vS1 of WT mice, but to a lesser extent in *Fmr1* KO mice. But some caveats to this analysis could not be addressed in the current study. First, we compared normalized weights between neurons, which may increase the contribution of certain neurons that could not be described well by the linear model. We did, however, use cross-validation to discover the best hyperparameter for each model and included only animals that had good fits using these regressors. Second, we grouped neurons across animals to compare between genotypes; while this approach is useful for highlighting differences in average weights between groups, it is blind to neuron-specific differences. Future analyses will focus on unique explained variance[59] and decoding accuracy (like what we report for locomotion—Supplementary Fig. 7c). This will be useful to isolate unique contributions of regressors to neural activity and to investigate genotype differences in the encoding of orbital area and social context.

Unlike vS1, activity in the BLA showed no preference for social or object touch during voluntary interactions. This was further supported by the linear encoding model findings, which showed lower values for normalized $\beta$ weights for context in BLA neurons than in vS1 neurons. Thus, the social valence of a stimulus does not seem to matter to BLA neurons, at least in our assay. This was surprising considering the roles that have been ascribed to BLA in social exploration and approach[25,46]. Instead, BLA neurons showed a preference for the most aversive stimulus, forced object touch (and a greater percentage of object-preferring cells) in mice of both genotypes. We interpret this to mean that touch within personal space matters to the BLA, and that aversive stimuli are represented in amygdalar circuits[26,60]. The BLA does receive projections from the ACCx and has been linked to learning of aversive sensory stimuli[26,29] and with the encoding of exploratory and aversive behavioral states[24].

Activity in the tSTR was modulated in opposite ways by social vs. object touch, depending on whether the interaction was forced or voluntary. This implies that neurons in the tSTR, unlike those in vS1, care less about stimulus context (social valence) and more about whether the interaction takes place within personal space (i.e., no choice but to interact) and about the ultimate behavioral response. Indeed, tSTR neurons were uniquely modulated during bouts of AFEs. Interestingly, tSTR receives inputs from amygdalar nuclei and prefrontal cortex[46,61].

During voluntary interactions, decoders trained on DLC labels showed a significant increase in accuracy upon touch in WT mice, but not *Fmr1* KO mice. This suggests that Fragile X mice display similar levels of aversion to both social and object presentations, and that they

fail to recognize the social valence of a visitor mouse. This is consistent with the social disinterest that others have reported for *Fmr1* KO mice using a social interaction test[62]. Furthermore, we find that having a choice to engage with another mouse matters, and that *Fmr1* KO mice (but not WT mice) find forced social touch at close proximity more aversive compared to when they voluntarily initiate social touch from a distance. Thus, social interactions that invade personal space may be especially bothersome in autism. This is perhaps the first demonstration of behavioral responses related to peri-personal space, a topic of increasing interest in the autism field[60,63,64]; but additional studies will be needed to investigate the neural basis of why invasion of one's personal space can be especially aversive. Future studies could also compare differences in behavioral responses (and neural activity) for different social scenarios, such as responses to familiar vs. stranger mice (including interactions that assess empathy), or responses of a female dam to one of its pups.

Cortical and striatal circuits of *Fmr1* KO mice, unlike those of WT mice, were not preferentially modulated by either social or non-social stimuli during voluntary interactions (also reflected in the reduced decoding accuracy for touch context based on neural activity). We conclude that a reduced ability to discriminate social from non-social touch at the circuit level could explain the reduced social interest in autism. We previously demonstrated that the maternal immune activation model of autism shows similar aversion to social touch as *Fmr1* KO mice[21]. It will be important to investigate how cortical, striatal, and amygdalar circuits in this and other models respond to social touch. Because WT mice showed less aversion to social touch, it would also be interesting to record from the NAc and related circuits implicated in pleasure/reward. Moreover, chemogenetic or optogenetic manipulations within these regions would help establish causal links between these circuits and social avoidance, which could be important for the development of strategies that alleviate social deficits in autism.

## Methods

### Animals
Adult male and female C5BL/6 mice at postnatal day 60–90 were used for all experiments. A cohort of adult mice (nine males and eight females) was used for TRAP labeling of neurons activated by social/object touch. These so-called "TRAP" mice were obtained by crossing Fos$^{2A-iCreER/+}$ (TRAP2) (JAX line 021882) with R26$^{Ai14/+}$ (Ai14) (JAX line 030323).

A second cohort of mice (>20 g in weight) was used for electrophysiological recordings and was derived from the following mouse lines based on prior publications: WT B6J (JAX line 000664), *Fmr1* KO (JAX line 003025)[32,33,65–67]. In total, nine WT (six males and three females) and *Fmr1* KO mice (six males and four females) were used for Neuropixels recordings.

All mice were group-housed with access to food and water using HydroGel (ClearH$_2$O) *ad libitum* under a 12-h light cycle (12 h light/12 h dark) in controlled temperature conditions. Mice with Neuropixels implants were single-housed during habituation and behavioral testing to avoid damage to the implant by group housing with other animals (~1–2 weeks). All experiments were done in the light cycle and followed the U.S. National Institutes of Health guidelines for animal research under an animal use protocol (ARC #2007-035) approved by the Chancellor's Animal Research Committee and Office for Animal Research Oversight at the University of California, Los Angeles.

### TRAP2 mice drug preparation
4-hydroxytamoxifen (4-OHT, Sigma Aldrich #H6278) was dissolved in 20 mg/mL in ethanol and was aliquoted and stored at −20 °C up to 4 weeks. On the day before behavioral testing, 4-OHT was redissolved in 70% EtOH by warming the aliquot at 37 °C and vortexing vigorously for 1 min 2–3 times. Corn oil (Sigma Aldrich, C8267) was added to each aliquot for a final concentration of 10 mg/mL of 4-OHT. The aliquots were vacuum centrifuged for 30 min until all EtOH was evaporated. 4-OHT was stored at 4 °C for next-day use.

### Behavioral social/object touch experiments for TRAP labeling
To identify brain regions involved in social touch, we used the behavioral assay we recently developed[21]. Adult TRAP2 mice (and the corresponding visitor mice in the social touch assay) were first surgically implanted with a titanium head bar. Briefly, mice were anesthetized with isoflurane (5% induction, 1.5–2% maintenance via nose cone v/v) and secured on a motorized stereotaxic frame (Kopf; StereoDrive, Neurostar) via metal ear bars. The head of the animal was shaved with an electric razor, and the skin overlying the skull was then sterilized with three alternating swabs of 70% EtOH and betadine. A 1 cm long midline scalp incision was made with a scalpel, and the custom U-shaped head bar (3.15 mm wide × 10 mm long) was secured on the back of the skull first with Krazy Glue and then with a thin layer of C&B Metabond (Parkell) applied to the dry skull surface. The entire skull was then covered with acrylic dental cement (Lang Dental). This surgery lasted ~15–20 min, and mice fully recovered within 30 min, after which they were returned to group-housed cages.

At least 48 h after head bar implantation, TRAP2 mice were habituated to head-restraint, to running on an air-suspended 200 mm polystyrene ball, and to the movement of a motorized stage that was used for repeated presentations of an inanimate object or a stranger mouse. The stage consisted of an aluminum breadboard (15 × 7.6 × 1 cm) attached to a translational motor (Zaber Technologies, X-LSM100A), the movement of which was fully controlled through MATLAB (Mathworks). All of this occurred in a custom-built, sound-attenuated behavioral rig (93 × 93 × 57 cm) that was dimly illuminated by two infrared lights (Bosch, 850 nm). For habituation of TRAP2 mice, test mice were placed on the ball for 20 min each day for 14 consecutive days before testing. In parallel, "visitor" mice (strangers to the test mouse) were habituated to head-restraint in a plexiglass tube (diameter: 4 cm) on the motorized stage. The stage translated at a constant speed of 1.65 cm/s. In its neutral starting position, the snout of the visitor mouse was 6 cm away from that of the test mouse.

Following habituation, all TRAP2 test mice were single-housed the day before the social touch assay[21]. On the day of behavioral testing and 30 min prior to testing, TRAP2 mice were injected with 4-OHT (50 mg/kg, i.p.). TRAP2 test mice were tested under three different conditions: 1. no touch, in which the platform moved back and forth in repeated bouts but was empty; 2. object touch, in which test mice experienced repeated bouts of forced interactions with a plastic 50 mL Falcon conical tube; and 3. social touch, in which test mice experienced repeated bouts of forced interactions with a visitor novel mouse (stranger to the test mouse). For the forced object/social interactions, the stage stopped at a position that brought the tip of the plastic tube or the snout of the visitor mouse in direct contact with the snout of the test mouse. These positions were calibrated before each experiment. For the condition where the platform was empty, the stage was moved to a set template position that was tested during calibration for forced interactions. Each bout lasted 5 s, with a 5 s ISI during which the platform moved away by 1 cm and the object/mouse was out of reach of the test mouse's whiskers. The ISI included a 1.2 s period of back-and-forth travel time for the platform. Each session (no touch, social touch, object touch) lasted 30 min, which was equivalent to a total of 180 such presentations.

Following the assay, TRAP2 test mice were returned to their single cage housing until the end of the day (~6–8 h), at which point they were placed back in group housing, and then they were perfused at 72 h. For each session of behavioral testing, at least three mice were used (one mouse each for the no touch, object touch, and social touch conditions).

## Histology and quantification of cFos expression in TRAP2 mice

Seventy-two hours after 4-OHT induction (to allow *Cre* recombination to occur), TRAP2 mice were transcardially perfused with 4% paraformaldehyde (PF) in cold PBS (0.1 M), and their brains were harvested and left overnight in 4% PF. Next, fixed brains were sliced coronally to obtain 60 µm sections. The coronal sections were mounted on VectaShield glass slides and stained with DAPI (Vector Laboratories). Sections were imaged on an Apotome2 microscope (Zeiss; 10× objective). Images were taken as a z-stack ranging from 30 to 50 µm (Zen2 software, Zeiss). ImageJ was used to quantify the density of cells expressing tdTom in each brain region (cFos-tdTomato$^+$ cells/mm$^2$). Cell densities in each brain region from "object touch" mice (TRAP2-OBJECT) and "social touch" mice (TRAP2-SOCIAL) were normalized to averaged cell density in each brain region from "no touch" mice (TRAP2-CTRL) in the same session of behavioral testing to account for variability in tamoxifen preparation from one session to the next.

## Surgical implantation of Neuropixels probes for chronic recordings

Each Neuropixels 1.0 probe (Imec, PRB_1_4_0480_1_C) was first connected to the acquisition hardware to confirm that the probe was functional using the SpikeGLX data acquisition software (see below) both prior to and after soldering a grounding wire (0.01 inches, A.M. Systems) to the ground and reference pads in the probe flex cable. The probe was inserted and screwed into a dovetail probe holder (Imec, HOLDER_1000_C) and set aside for surgical implantation. A custom-made external chassis cover (eventually used to protect the probe during implantation) was 3D-printed (Hubs) using standard black resin (Formlabs, RS-F2-GPBK-04). The CAD files for the 3D-printed cover were acquired at https://github.com/Brody-Lab/chronic_neuropixels[68]—we used the external casing part.

Adult mice were anesthetized with isoflurane and placed on a motorized stereotaxic frame. Their head was shaved, and the scalp sterilized as above. A 1 cm long midline scalp incision was made with a scalpel and a small burr hole (0.5–0.8 mm diameter) was drilled over the cerebellum (2 mm posterior to Lambda) with a dental drill (Midwest Tradition) through which a stainless-steel ground screw (M1, McMaster Carr) was loosely screwed. A second burr hole (0.5–1 mm diameter) was drilled at the probe implantation site at coordinates −1.46 AP, 2.9 ML 3.75 DV (in mm). This allowed for targeting of vS1, tSTR, and BLA simultaneously with a single-shank Neuropixels probe. Saline-soaked Surgifoam (Ethicon) was placed in both craniotomies while a thin layer of C&B Metabond (Parkell) was applied to the dry skull surface. A small well (0.75 cm diameter, 1 cm height) was built around the craniotomy site with self-adhesive resin cement (RelyX Unicem 2 Automix, 3M ESPE) and set with a dental curing lamp (Sino Dental). Surgifoam was removed from the implantation craniotomy, and saline was applied to maintain tissue hydration.

Before insertion, the probe holder (with the probe attached) was screwed and secured to a micromanipulator (StereoDrive, Neurostar). To enable histological reconstruction of the probe tract, the tip of the shank was dipped for 30 s in DiI (in 1–2 mg/mL in isopropyl alcohol; Sigma Aldrich, applied onto Parafilm, Bemis). DiI fluorescence enabled subsequent histological reconstruction of the probe tract in fixed tissue sections. The ground wire soldered to the probe was then wrapped around the ground screw, as it was being screwed into the ground burr hole. Conductive epoxy (8331, MG Chemical) was applied on the ground screw and wire. A titanium U-shaped head bar (3.15 × 10 mm) was affixed to the skull with Metabond (Parkell) caudal to the ground screw, to allow for head-restraint during recordings and behavioral testing. Next, the probe shank was lowered at a rate of 10 µm/s with StereoDrive through the implantation craniotomy. Saline in the cement well was then absorbed carefully with Surgifoam and replaced with Dura-Gel (Cambridge Neurotech). Additional light-curable resin

cement was applied and cured to the cement well and onto the probe base (avoiding contact with the shank). An additional layer of Metabond was applied on the skull, including on the ground craniotomy site and along the outside of the resin cement well. The two parts of the external chassis were then placed around the probe and cemented together and to the resin cement wall with acrylic (Lang Dental). This surgery lasts 3–4 h, and hydration was provided by injecting saline every hour (0.1 mL, i.p.). Mice fully recovered within 1–2 h after surgery. Afterwards, implanted animals were single-housed for habituation and behavioral testing. Mice were given carprofen (Rimadyl) immediately after surgery and again at 24 and 48 h post-op, and given ad lib access to HydroGel (ClearH$_2$O) and food.

## Social touch assay in mice with chronic Neuropixels implants

Following probe implantation, test mice were subjected to the social touch assay described above, but in addition to forced object/social touch, we introduced additional interactions (see below). First, mice bearing Neuropixels implants were habituated to head-restraint, to running on the polystyrene ball, and to the behavioral apparatus (just as for the TRAP experiments above, but for only 7–9 days).

Following habituation, test mice were subjected to both voluntary and forced interactions with a visitor mouse or a novel inanimate object over the course of 2 days. On day 1, test mice were placed on the ball and recorded for a 2 min baseline period (the plexiglass tube on the moving stage was empty). Next, we inserted the plastic object (50 mL Falcon conical tube) into the plexiglass tube on the motorized stage. For this control interaction, the test mouse first experienced a 2 min period of no touch but was able to visualize the object in the neutral position (before touch, 6 cm away). Next, the motorized stage moved the object to within whisker reach of the test mouse for a total of 40 such presentations of either voluntary (whisker-to-object) or forced (snout-to-object) object touch. Each bout lasted 5 s, with a 5 s ISI during which the platform moved away by 1 cm and the object was out of reach of the test mouse. The ISI included the total travel time of the platform, 1.2 s.

After this object touch session, the test mouse was returned to its cage to rest for at least 1 h before being head-restrained again to undergo either voluntary or forced social touch session (same type of touch as previous session for object touch) with a visitor mouse. A same-sex, same age (P60-90) novel WT mouse was head-restrained inside the plexiglass tube on the motorized stage. Following a 2 min period in the neutral position where the test mouse could see but not touch the stranger mouse, the motorized stage moved to the position for voluntary social touch (whisker-to-whisker) or forced social touch (snout-to-snout) for 40 bouts (also lasting 5 s with a 5 s ISI where the mouse on the platform moved out of reach of the test mouse). The test mouse was then returned to its cage for at least 24 h.

On day #2 of behavior testing, the mouse was placed back on the ball again for a 2 min baseline period followed by a 2 min period of no touch. Depending on what interaction the test mouse had received, voluntary or forced object and social touch on testing day #1, the mouse received 40 presentations of the alternate touch context with a novel object and another stranger mouse.

## Electrophysiological recordings

During the social touch behavioral assay and tactile defensiveness assay, electrophysiological recordings were performed using Neuropixels 1.0 acquisition hardware (Imec). The acquisition hardware was used in combination with PCI eXtensions for Instrumentation (PXI) hardware (PXIe-1071 chassis, PXIe-8381 remote control module, and PXIe-6341 I/O module for recording analog and digital inputs, National Instruments). SpikeGLX software was used to acquire data (https://github.com/billkarsh/SpikeGLX, HHMI/Janelia Research Campus). Recording channels acquired electrical signals from the most dorsal region of the vibrissal primary somatosensory cortex (vS1) down to the

most ventral region of the BLA using the deepest of the 960 electrode sites. Electrophysiological signals were processed with Kilosort2.5 (https://github.com/MouseLand/Kilosort) using default parameters for spike sorting and then manually curated with Phy2 (https://github.com/cortex-lab/phy)[69]. Only well-isolated single units were used for electrophysiological data analysis. After manual curation, we selected units that passed the following criteria: ISI violation <10%, amplitude cutoff <10%, and median amplitude >50 μV[70].

### Removal of Neuropixels probes
Neuropixels probes were explanted for subsequent re-use. Mice implanted with Neuropixels were anesthetized with isoflurane and secured on a stereotaxic frame. The external chassis was removed with a dental drill, and any excess acrylic or resin cement around the probe dovetail was gently drilled off while avoiding direct contact with the probe. The dovetail holder was inserted and screwed into the probe and attached to the stereotaxic arm. Resin cement was carefully drilled around the circumference of the resin cement well to separate the skull from the probe. Once the skull and probe were separated, the probe was lifted up using the motorized micromanipulator until the probe was completely outside the resin cement well. The probe was removed from the dovetail holder, and forceps were used to gently remove any excess resin from the probe. After removal, the probe shank was fully immersed in 1% tergazyme (Alconox) for 24–48 h, followed by a 1–2 h rinse in distilled water.

### Histology and fluorescence imaging of probe location
Following probe removal, mice were anesthetized with 5% isoflurane and transcardially perfused with 4% PF and post-fixed overnight. At least 24 h after perfusion, the fixed brain was rinsed with PBS and sliced coronally with a vibratome to generate 50 μm sections. The coronal sections are mounted on slides with VectaShield mounting medium (Vector Laboratories). DiI fluorescence in each section per brain was imaged on an Apotome2 microscope (Zen2 software, Zeiss; 5× objective, 5 × 5 grid of images). ImageJ was used to visualize each section image and reconstruct the entry point of the probe shank to the tip of the probe shank in the brain.

### Electrophysiological data analysis
We first converted action potential spikes to firing rate estimates (in spikes per second) for each single unit by binning spike counts in 50 ms bins and dividing by the bin size. To generate peristimulus time histograms (PSTHs), we smoothed firing rates with a 250 ms moving window and then averaged all touch presentations from 2 s before the onset of touch to 2 s after the end of touch [−2 to +7 s]. Units were assigned as belonging to vS1, tSTR, or BLA based on the dynamics of their action potential spiking by depth and across time (Supplementary Fig. 4). Regular-spiking (RS) units in vS1 were identified based on the duration of their spike waveform (≥400 μs peak to trough).

### Classification of single-unit responses to social and object touch
Despite the heterogeneity of single unit responses to voluntary and forced touch, we sought to determine whether some units behaved similarly to others, i.e., whether there exist different functional groups of neurons in each brain region. We performed clustering of single units twice using PSTHs of all presentations (object and social) of (1) voluntary touch and (2) forced touch. Clustering of the PSTHs was also done separately for each brain region (vS1, tSTR, and BLA), so the procedure was employed 6 times. The clustering procedure we used takes the z-scored, trial-averaged PSTH of each unit and combines all responses into a matrix (PSTH × unit). Units from WT and *Fmr1* KO mice were included together within the PSTH x unit matrix. PCA was then performed on this matrix, followed by k-means clustering of the top *k* components that explained >95% of the variance. The gap statistic criterion was used to estimate the ideal number of clusters for

each clustering, followed by visual inspection of the temporal firing of units in each cluster (to confirm their different responses). We applied 10,000 iterations of k-means clustering for each clustering procedure performed, each with a new initialization of the centroids to validate the appropriate cluster assignment of a unit. Clustering of single units was also performed on vS1, tSTR, and BLA units separately for the recording session in which the test mouse received forced touch from an inanimate toy mouse and similarly for the session in which the test mouse received forced touch from an anaesthetized mouse. The PSTHs were split by brain region (vS1, tSTR, or BLA) and grouped as either voluntary (average of all trials of voluntary object and social touch) or forced (average of all trials of forced object and social touch). By grouping units in this manner, we could compare how a unit assigned to a cluster by k-means responds differentially to voluntary object and social touch and responds differentially to forced object and social touch.

### Modulation of single units by social and object touch
To quantify differences in the mean firing rates of units in each cluster between voluntary or forced object and social touch, we calculated the z-score firing rate normalized to the average firing rate during the ISI period. To assess the modulation of units by social and object touch, we grouped together neurons from clusters with similar temporal properties. Units in clusters that were moderately to strongly excited were grouped together ("excited" cells), as were moderately or strongly suppressed units ("suppressed" cells). A modulation index (MI) was used to calculate how much the firing rate (FR) of each unit changed during the stimulus period (stim) relative to the ISI period in each trial: Eq. (1)

$$MI = \frac{FR^{stim} - FR^{ISI}}{FR^{stim} + FR^{ISI}} \quad (1)$$

The MI was calculated using three different time ranges for the stim and ISI period. For calculating the MI over the entire stimulation period ($MI^{STIM}$), we used the mean FR over 5 s during which the platform was stopped (touch) for $FR^{stim}$ and the mean firing rate over the 5 s ISI for $FR^{ISI}$. For the MI of the first few seconds of the presentation period ($MI^{SHORTSTIM}$), we used the mean FR from the first 3 s of presentation for $FR^{stim}$ and the mean FR for the 3 s prior to the presentation onset for $FR^{ISI}$. For the MI during the period that the platform moves ($MI^{PLATFORM}$), with the platform as the stim, we used the mean FR from [−1 1] s with time 0 as the presentation onset $FR^{stim}$ and the mean FR from [−3 −1] s for $FR^{ISI}$. $MI^{STIM}$ was used to compare the modulation of vS1 suppressed and excited cells, tSTR excited cells, and BLA excited cells. We also assessed the MI of units in each cluster. For assessing modulation by cluster, $MI^{PLATFORM}$ was used for units in clusters that showed a change in FR during the initial onset of touch, $MI^{STIM}$ was used for units in clusters that showed a sustained change in FR during the period of touch, and $MI^{SHORTSTIM}$ was used for units in clusters that showed a larger change in FR during the initial onset of touch as well as a sustained change in FR during touch.

### Single neuron coding of stimulus preference and behavior
To determine which units show a clear preference for object vs. social touch, we used ROC analysis, which was applied to the firing rate (Hz) during the presentation period [0, 5 s][71–73]. Each unit's preference was calculated based on the firing rate response to each trial relative to the mean PSTHs for object touch and social touch trials. Each trial was assigned a decision variable (DV) score, and the DV for social touch and object touch trials was calculated as follows: Eqs. (2) and (3)

$$DV^{social} = t_i((meanSocial)\_(k \neq i) - meanObject) \quad (2)$$

$$DV^{object} = t_i(meanSocial - (meanObject)\_(k \neq i)) \qquad (3)$$

where $t_i$ is the firing rate for the current ($i$th trial) and *meanSocial* and *meanObject* correspond to the mean social and object touch PSTHs. An ROC curve was obtained by varying the criterion value for the DV, and the area under the ROC (auROC) was calculated from the ROC curve using the MATLAB function *trapz*. The auROC value was considered significant by bootstrapping 1000 times with a threshold probability of 0.05. Single units that were excited by touch and showed significant auROC values >0.5 were deemed to show preference for social touch (social cells), and those with significant values <0.5 showed preference for object touch (object cells). For suppressed units, those with significant auROC values <0.5 were social cells, and those with significant values >0.5 were object cells. Units with non-significant auROC values were considered as showing no preference.

## Decoding touch context from neuronal activity and behavior

We used SVM linear classifiers (default parameters of fitcsvm in MATLAB 2023b, radial basis function kernel with box constraint parameter = 1 for every classifier and 10-fold cross-validation) to determine how well all neurons, or neurons within a particular cluster, could decode touch context (object vs. social presentations) under voluntary or forced conditions. We used activity from 80% of trials (64/80 of both object and social touch trials) as the training dataset, and the remaining 20% (26/80) was used for testing the classifier's accuracy. Firing rates, either as an average of the stimulus period (0 to +5 s) or binned as 50 ms across the stimulus period (−2 to +7 s), were used as the feature space of the SVM. 100 iterations of the decoding analysis were performed in which the neurons of a cluster and trials were randomly chosen for the training and test datasets. The mean decoding accuracy was calculated based on the average performance of all 100 iterations. In addition, we separately trained the decoder on neural data in which the trial labels were randomly shuffled for the test dataset ("Shuffled").

We also tested decoding accuracy after excluding neurons with large changes in baseline firing between object and social touch presentations occurring on the same day. Neurons were excluded from being used in the linear classifier if their baseline firing in the second presentation session was greater or smaller than 1.5 standard deviations of the mean baseline firing in the first session.

To determine if decoder performance is influenced by an animal's locomotion levels, we compared decoder accuracy between animals that consistently ran during all presentations of social and object touch on a given day. Furthermore, we tested decoder performance if we included locomotion (running motion energy binned at 50 ms), or a shuffled version, in the feature space of the SVM.

Decoding was performed with neurons within each cluster by brain region. A different population size of neurons, ranging from 1 neuron to 20 neurons, was used for decoding context from averaged activity during the stimulus period, and 20 neurons were used for decoding context across time. For decoding context (object vs social) or choice (voluntary vs forced) from behavior across time, we binned each behavioral measure in 100 ms bins. For decoding context from facial motion, a total of 23 DLC labels were used, and running avoidance, eye area, saccade direction, and whisker protraction for decoding context from avoidance and aversive behaviors (see analysis of behavior below).

## Data analysis of behavioral data

During the social touch behavioral assay and tactile defensiveness assay, high-resolution videos (.avi files) were recorded of the test mouse's eye, face, and body using three cameras (Teledyne Flir, Blackfly S USB3) at 120 FPS for behavioral analyses. Locomotion and running direction, facial expressions (including AFEs), and pupil saccades were analyzed from these videos of the eye, face, and body. Running direction and AFEs (orbital tightening and whisker protraction) were quantified using MATLAB DLC[21,35,74]. For locomotion, Face-Map was used to track the motion energy of a polystyrene ball as the animal was running[35]. For analysis of pupil saccades and facial motion, a DLC neural network was trained on images from the eye or face videos to identify markers on the mouse's pupil, eye, six whisker follicles, mouth, and nose. The markers on the animal's eye or face were used to quantify the following metrics: motion energy of each marker (change in marker position every two frames), saccades along temporal-nasal plane (displacement of pupil on the $x$-axis), and eye area (pixel area of eye markers)[75]. For analysis of pupil saccades and facial motion and expressions (including AFEs), we excluded video frames when the animal was blinking, grooming, or other movements obscured the animal's face.

## Linear encoding model

A linear encoding model was constructed for each animal/session to predict average trial-to-trial neural activity from a set of stimulus parameters and behavioral variables. The model was generated by combining multiple stimulus and behavioral variables into a design matrix (Fig. 7a). Variables, denoted as regressors (a total of 13), were either discrete or continuous and either designated the stimulus context, aversive behaviors (running avoidance, orbital tightening and whisker protraction) or other behaviors (motion energies of all markers on animal's eye and face, pupil size and saccade direction, running motion energy and locomotion). The average value for each regressor was taken over the course of each stimulus presentation (5 s period), such that each regressor has 80 timepoints in the design matrix (13 regressors × 80 trials) corresponding to both forced object and social touch stimulus presentations. A separate matrix consisted of the average firing rate of each neuron per stimulus presentation (# neurons × 80 trials) for all forced object and social touch presentations. To minimize overfitting, we input both the design matrix and neural data matrix into a ridge regression, linear encoding model. Each model generated the $\beta$ weights of each regressor for every neuron and the overall explained variance for a given animal. If the overall explained variance ($cvR^2$) was negative, this denoted that the model poorly fit the neural data and that the animal was excluded from further analysis. The $\beta$ weights for each neuron were normalized to the sum of the $\beta$ weights for all 13 regressors from that neuron. Normalized $\beta$ weights were used to compare which regressors contributed the most to neuronal activity across WT and *Fmr1* KO mice.

## Statistical analyses

Statistical tests were performed in Prism software (GraphPad). Statistical analyses of normality (Lilliefors and Shapiro−Wilk tests) were performed on each dataset; if data deviated from normality ($p < 0.05$) or not ($p > 0.05$), appropriate non-parametric and parametric tests were performed. For parametric two-group comparisons, a Student's $t$-test (paired or unpaired) was used. For non-parametric tests, we used the Mann−Whitney test (two groups) and the Kruskal−Wallis test (repeated measures). Multiple comparisons across touch conditions and genotypes/groups were analyzed using two-way ANOVA with post-hoc Bonferroni's test. If the data were non-normal, we applied a logarithmic transformation to the data and compared the two-way ANOVA with and without the transformation. Since the statistical output of the two-way ANOVA was similar for the transformed and the non-transformed (non-normal data), we used the latter. Statistical significance was set at $p < 0.05$. However, we do point out interesting trends that approached significance in Fig. 1d ($p = 0.051$), Fig. 3d ($p = 0.055$), Fig. 5c ($p = 0.054$), and Supplementary Fig. 10c ($p = 0.052$).

All experiments were conducted on animals from at least two different litters for each genotype/group. For the figures related to the

TRAP experiments, we used the number of tissue sections (with the same number per animal) as the sample size. For the remaining figures, the statistics were done on either the number of units as the sample size or using individual mice as the sample size (averaged over cells for different mice), superimposed on individual data points. Because there are important sex differences in both the prevalence and symptoms of ASD[76,77], we distinguished males from females across all figures (squares depict males, circles depict females). In all figures, the error bars denote the standard error of the mean (SEM). Robust regression and outlier removal (ROUT) analysis was used to exclude outliers for data represented as individual mice. The standard deviation rule was used to exclude neurons with large changes in baseline firing and neurons with large $\beta$ weights in the linear encoding model for a given animal. One WT animal was excluded from behavioral decoding as videos were not synchronized across cameras.

A summary of statistical analyses for each figure is provided in Supplementary Table 1.

### Reporting summary
Further information on research design is available in the Nature Portfolio Reporting Summary linked to this article.

## Data availability
The data analyzed for this study have been deposited in FigShare. For electrophysiological data, see https://doi.org/10.6084/m9.figshare.28689119.v1. For behavioral data by trial, see https://doi.org/10.6084/m9.figshare.28689122.v1. For TRAP data, see https://doi.org/10.6084/m9.figshare.28689209.v3. The data generated in this study is also provided in a Source Data file for each figure. Source data are provided with this paper.

## Code availability
Custom code written in MATLAB and Python for analysis of electrophysiological neural data and behavior is available at https://github.com/porteralab and adapted from code in https://github.com/jcouto/pnc_spks.

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

## Acknowledgements

We are grateful to Weizhe Hong, Lukas Oesch, and Anne Churchland for advice on the analysis of behavioral and neural data. We thank Will Zeiger and Sotiris Masmanidis for comments on the manuscript. Kimberly Battista (https://www.battistaillustration.com) made the illustration in Fig. 1a. This work was supported by the following grants: R01NS117597 (NIH-NINDS), R01HD108370 and R01HD054453 (NIH-NICHD), Department of Defense (DOD, 13196175) awarded to C.P.-C., Training in Neurotechnology Translation T32NS115753 (NIH), F31HD108043 (NIH/NICHD), and a graduate student fellowship from the Achievement Rewards for College Scientists Foundation to T.C., and the

CARE Fellows Program to A.H. Some figure panel cartoons were generated with BioRender.

## Author contributions

T.C. and C.P.-C. conceived the project. T.C. performed the experiments and analyzed the data with help from A.H. and J.C. T.C. and C.P.-C. prepared the figures and wrote the paper with input from other authors.

## Competing interests

The authors declare no competing interests.
