## [Peer Review file · Nature Communications]

A reduced ability to discriminate social from non-social touch at the circuit level may underlie social avoidance in autism

Corresponding Author: Professor Carlos Portera-Cailliau

Version 0:

Reviewer comments:

Reviewer #1

(Remarks to the Author)

The study investigates the neural circuit mechanisms underlying the processing of social from non-social touch and how it's related to social interaction in Fmr1 KO mice, the model of Fragile X Syndrome, the most common form of autism. The authors have developed a novel social touch assay. While using this assay together with Neuropixels probes, they perform recordings in three brain areas: the vibrissal somatosensory cortex (vS1), the striatum (tStr) and basolateral amygdala (BLA). The authors describe neural correlates of social and non-social touch responses in voluntary and forced paradigms. Fmr1 KO mice demonstrate aversion to both social and non-social interactions. This observation also correlates with impaired encoding of social valence at the neural circuit level. WT mice demonstrated aversion to object touch but not to social touch, while Fmr1 KO mice showed aversion to both, indicating a potential neural basis for social avoidance in autism. This manuscript is very extensive. The work is innovative and original, the methodology is rigorous with multiple controls and different characterizations.

There are some questions related to the manuscript:

- 1) It would be interesting to see if there is a statistical difference between the neural responses of different brain areas during social and object touch between WT and Fmr1 KO mice. It would be interesting to test if there a specific brain area responsible for this difference, too.
- 2) The decoding accuracy of 60% doesn't seem to be very high. Does the accuracy of decoding increase with more cells or longer time? Is the high accuracy of decoding for the clusters excited by touch explained by the increased number of spikes? What about the clusters inhibited by spikes?
- 3) One of the important questions related to the manuscript is: are the same neurons encoding information about social touch also encoding information about object touch? Are these neuronal populations overlapping or not? Is the information encoded at the level of synapses or individual neurons?
- 4) How is this information encoding impaired in Fmr1 KO mice?

Minor comments:

- 1) In Supplemental Figure 10, is there a statistical difference between the modulation indices of WT and Fmr1 KO mice in social compared to object touches? There seems to be a potential difference between the suppressed, but not excited units? Is this the case and if yes, what is the interpretation of this observation?
- 2) The description of the experimental paradigm for the voluntary touch vs forced touch is not very well explained.

Reviewer #2

(Remarks to the Author)

Chari et al quantified aversive facial movements in wild-type mice and in mice with the Fmr1 knockout, and evaluated the relationship between touch context and neural activity. The results are novel and interesting, and advances our understanding of social interactions in mice especially wrt Fmr1, however I think more analysis is required to verify the decoder results and to disentangle facial movements and social valence.

1. The decoders were trained to discriminate two trial types (object vs social) that occurred in two separate blocks. Unfortunately the design of the experiment means that slow drift in neural activity could result in high decoding accuracy. Could you exclude neurons which had changes in average firing rate across the two blocks and still accurately discriminate? Also, given locomotion was different, could you discriminate trials with similar locomotor activity? This would

check if the neurons/behaviors are coupled to locomotion. You could additionally build a decoder with locomotion+neural activity/whisking, and show that this decoder performs better than locomotion alone. Please also provide information about training the decoders, e.g. regularization coefficients (if different regularization coefficients were used for different sessions then a validation set is also necessary). Also, please try to define the decoder (e.g. object vs. social) in the figure panel titles (this may be less necessary when the captions are closer to the figures in the final paper).

2. Cluster analysis and sorting in Figure 3 should be performed in a cross-validated way: training trials used to find clusters/sorting, then average cluster activity plotted for test trials. This ensures that outlier trials do not drive the clustering. These test trials can then also be used to compute the modulation indices.

3. I am wondering if the fact that the neural activity is not discriminative of social vs object context in the Fmr1 mice is linked to the fact that the Fmr1 mice makes more aversive movements to both compared to wild-type. This is difficult to disentangle but perhaps the authors could see if there are neurons correlated with aversive movements in the population and see if they are potentially more differentially modulated.

Reviewer #3

(Remarks to the Author)

In "A failure to discriminate social from non-social touch at the circuit level may underlie social avoidance in autism," Chari et al. aim to link differential behavioral responses to specific touch contexts (social/non-social) and touch choices (voluntary/forced) to differential neural responses evoked by the same stimuli in three brain regions: the somatosensory cortex (for sensory processing), tail of striatum (sensorimotor decision-making), and basolateral amygdala (emotional processing). The authors compare sensory-evoked behavioral and neural responses in wild-type mice to those observed in mouse models of autism (Fmr1 KO) and describe reduced social/non-social discrimination in touch-evoked neural activity in Fmr1 KO mice that could potentially underlie the increased aversion to social touch seen in ASD.

The question posed by the authors is compelling and aligns well with the existing literature on altered social categorization in mouse models of autism. The experimental design is straightforward and generally well-suited to addressing the research questions, despite the inherent challenges of using head-fixed mice to study social responses. However, some inconsistencies in the presented data and certain analytical choices raise concerns about the authors' ability to fully support their main conclusion.

Major concerns:

1. Inconsistency between data and main claim: The paper's primary claim, as stated in the abstract and title, is that neurons in Fmr1 KO mice "do not discriminate social valence." However, the data presented seem to contradict this assertion. For instance, in Supplementary Figure 10, both during forced touch and voluntary touch, neurons recorded in Fmr1 KO mice are differentially modulated when presented with social versus object touch, indicating clear, significant discrimination between the stimuli. Specifically, both excited and suppressed units in vS1 of Fmr1 KO mice show more modulation for social versus object voluntary touch, and the same is true for tSTR and BLA during forced touch. This result alone challenges the paper's main claim.

Additionally, in Main Figure 6b, the authors present decoders for touch context (social vs. non-social) in wild-type versus Fmr1 KO mice, with mean decoder performance quantified in Fig 6c. For two out of the three regions tested, decoder performance does not differ between the genotypes. In the other region (BLA), the Fmr1 KO decoder performs worse than the wild-type, but still appears to be above chance levels, suggesting that some social discrimination is still apparent in this region for Fmr1 KO mice. The authors should clarify how these results support their claim, as the data currently appear to contradict the paper's central thesis.

2. Lack of direct correlation between behavioral and neural data: Another key claim of the paper is that differential neuronal responses underlie differential behavioral responses to touch stimuli. However, the authors do not directly correlate behavioral and neuronal data, except for a single panel (Fig. 3g). Instead, they analyze the behavioral response to a stimulus and the neuronal response to the same stimulus independently, leaving readers to infer a brain-behavior correlation that is not explicitly derived from the data. Since the authors recorded behavioral and neuronal responses simultaneously, if they wish to claim that specific neural responses underlie specific behavioral responses, they should present a direct correlation.

For example, can neuronal responses predict behavioral aversion levels on a trial-by-trial basis, rather than merely predicting stimulus context or choice? If aversion parameters are similar (regardless of stimulus context or choice), are the neural responses similar? Or, per animal—compared to some average or baseline modulation factor—when an animal's response to social touch is more aversive, is the neural response more similar to that evoked by an object? This approach would not only strengthen the paper's claims regarding genotype differences but also provide a more direct correlation between neuronal activity and aversion levels within each genotype. Overall, a manuscript that claims a relationship between behavior and neural activity should demonstrate this link explicitly.

3. Analytical choices: The authors make several analytical choices that could be reconsidered:

o Incomplete data presentation: The data presented in the figures do not always fully correspond to the experimental conditions defined in the paper (social/non-social, voluntary/forced, wild-type/KO), making it difficult to evaluate the consistency of the results. For example, Figure 1f, h panels are presented only for voluntary touch; Figure 1l (left) only for social touch; Figure 2 only for wild-type mice, not Fmr1 KO; Figure 3e only for wild-type, not KO; Figure 3g only for excited neurons; Figure 3h, i only for voluntary, not forced touch; Figure 5d only for voluntary; and Figure 6e only for forced touch, with no direct comparison to wild-type. Although the authors can choose which data to present in the main figures and which in the supplementary materials, the complete analysis of the datasets should be presented somewhere in the manuscript

(again, in the context of the chosen experimental groups) if the results are to be considered supportive of the conclusion.

- o Simplified neural data analysis: The authors' decision to pool together different clusters of units with distinct response patterns may obscure specific neuronal specializations that are relevant to the experimental questions. For example, in Figure 3e, cluster 9 and 10 are pooled together even though their response patterns differ substantially in terms of both dynamics and social preference. By averaging/pooling the clusters, the authors present results that are both not specific enough and could wash out neuronal specializations that are actually very relevant for the experimental questions. For instance, in cluster 10, the difference between social and non-social is much larger than in all other clusters—a finding that can be either muted for this cluster by pooling with other clusters, or increased overall specificity when averaged with clusters not showing such preference.
- o In Supplementary Figure 6, when the authors explore the social discrimination of neurons recorded in wild-type (wt) mice, only very few clusters of neurons (how many neurons overall is unclear) show differential responses to social versus nonsocial touch. These are pooled together with clusters that do not show significant preference to social valence, which implies that even in wt mice, the discriminability is not that prominent. This pooling may obscure important distinctions and undermine the specificity of the findings. The authors have many other options—such as using only clusters with the most differential modulation between social and non-social stimuli—that could provide a clearer understanding of the data. Can the authors explain their analytical choice?
- o Lack of data on stability, specificity and dynamics over time: The authors do not provide enough data to understand the stability or specificity of neural responses. For example, can neurons that are more modulated by the object voluntary condition (vs. object) be more modulated by the social forced condition (vs. object)? How stable is the neuronal modulation score? What about preference measurements over trials? Are these measures different across regions or genotypes? How do they correlate with the consistency and specificity of behavioral responses? What are the population sparseness and lifetime sparseness values? How do the dynamics of the different response types differ, and what is their correlation with dynamics of behavioral responses?
- o Insufficient Data on KO Mice: The authors also show substantially less data for the neural recordings conducted in KO mice. All the results presented for wild-type neuronal responses should also be presented for units recorded from KO mice.

4. Framing of the paper experimental design: Some aspects of the paper's framing seem somewhat arbitrary. For instance, "social" and "non-social" are broad categories and should not be generalized based on a singular stimulus. The authors test only one social and one non-social stimulus, even though previous work from the same group has shown differential responses to different types of social stimuli. It is possible that the observed results are related to other tactile properties of the stimuli rather than their social/non-social nature (e.g., the tube is smooth, the mouse is not; the tube is still, the mouse is moving). In that sense, "social vs. non-social" is really "mouse vs. tube." Additionally, the terms "voluntary" versus "forced" touch might need re-examination, as the distinction is not as clear-cut as implied. The concern is that "voluntary" versus "forced" is only "close but not touching" versus "touching your nose." These are not solely semantic distinctions but have a direct effect on the claim of the paper. The authors might consider adding more types of social and non-social stimuli or rethinking their definitions to solidify their arguments. Finally, a minor but noteworthy point is the use of the terms "personal space" and "aversion." It appears that all the stimuli presented in this study are aversive, at least based on the behavioral responses. Is this truly the case? The authors might consider including a pleasant or attractive stimulus to provide a broader perspective. Additionally, while the notion of "personal space" is frequently mentioned, it is neither empirically explored nor quantitatively defined in the study. It would be helpful to clarify the relevance of this concept to the paper's findings, or reconsider its inclusion if it does not add substantial value.

Other Concerns:

1. Decoding comparisons: The authors present various decoders, but many lack comparison with shuffled data, making it difficult to assess whether the decoding performance is above chance levels or differs between wild-type and Fmr1 KO mice. For example, in Figures 5d, 5g, and 6b, there is no comparison with shuffled data. The authors should ensure that decoding performance is always compared to chance and explore whether it exceeds it.

2. Clustering analysis: The clustering analysis of the neural data raises some questions. It is only presented for wild-type and not Fmr1 KO mice, and the justification for the number of clusters chosen is not clearly explained. The authors should provide more clarity on how clusters were identified and whether the same criteria apply to Fmr1 KO mice. Additionally, in Supplementary Figure 7, the authors discuss the "top-performing clusters" for decoding social context, but these clusters seem inconsistent with the data presented in the main figure 3. For example, in Supplementary Figure 7e, the authors claim that cluster 6 is most informative for forced touch, yet in Figure 3 and the main text, cluster 6 is defined as "unmodulated" by the stimulus social context. Similarly, cluster 14, rather than 15, seems to be more modulated by social context, although transiently. The authors should explain this inconsistency.

3. Statistical analysis: The statistical analysis and conclusions drawn from it should be revised for consistency.

- o Although the methods suggest that the authors apply reasonable statistical approaches to the data, throughout the paper, the authors do not treat the value of $\alpha < 0.05$ as a strict threshold. As a result, p-values larger than 0.05 are sometimes considered in the paper (implicitly or explicitly) as significant effects supporting the authors' claim. The main issue here is that in such cases, the authors have the discretion to consider the effect relevant or not—a situation that the setting of a statistical threshold aims to avoid. The authors should determine their threshold and adhere to it throughout the paper when discussing their results.
- o The results of many of the statistical analyses are not fully presented; only the post-hoc t-tests are shown. It is crucial to determine whether factors such as social context and genotype show a main effect or interaction effect before considering the post-hoc comparisons. Some of the post-hoc comparison results are also unreported. The authors should clearly present all statistical analysis results to ensure transparency and comprehensiveness.

o Additionally, any conclusions regarding differences between wild-type and Fmr1 KO mice should be drawn from direct comparisons between the genotypes, rather than from independent analyses of each. Consistency in this approach across the manuscript would strengthen the validity of the findings.

4. Importance of platform movement: The authors dedicate some analysis to "before touch" and "after touch" decodability. However, the importance of this analysis is not entirely clear. Isn't it expected that decodability of touch would increase once touch is initiated? If the key finding is that this phenomenon differs in Fmr1 KO mice, the authors should emphasize this specific comparison and discuss its implications in greater detail.

Conclusion:

This study addresses an important question in autism research and employs a suitable experimental approach. The authors' investigation of the neural basis for differential responses to social and non-social touch in wild-type and Fmr1 KO mice has the potential to provide valuable insights into ASD-related social avoidance behaviors. However, the inconsistencies between the data and the main conclusions, along with some analytical choices, currently limit the strength of the findings.

Version 1:

Reviewer comments:

Reviewer #1

(Remarks to the Author)

The manuscript is improved.

The revision is very detailed and thoughtful. All my concerns have been addressed.

Reviewer #2

(Remarks to the Author)

Thanks to the authors for thoroughly addressing my concerns, especially with respect to the behavioral controls. This is a valuable manuscript with novel findings on an important topic.

Reviewer #3

(Remarks to the Author)

I thank the authors for thoughtfully and thoroughly considering my comments. I agree that the revised manuscript provides the readers with additional important data, clarifying most of my comments. I also appreciate the authors' willingness to modify the title, abstract and text to address my main concerns. I, too, think that "a 50 cc polypropylene Falcon conical centrifuge tube" is perhaps too specific, but I appreciate the authors for acknowledging that directly addressing the complexity of the results, the inherent reductionism in choosing one stimulus to represent a category and the nuances in result interpretation strengthen the contribution of the manuscript to our understanding of the neural changes that might underlie ASD-related behavioral alterations.

I also very much appreciate the new figure presented by the authors (Fig. 7) and consider it a meaningful contribution to the manuscript. Since the authors themselves acknowledge that these are only preliminary results from a more comprehensive follow-up exploration (and indeed, measurements of explained variance will add much to this analysis), I have only a few comments, most of them similar in nature to my previous feedback: 1) In Fig. 7b, the distribution of β weights clearly shows that the activity of the selected units reflects contributions from several regressors. The authors should reconsider the use of the term "specific" here. 2) Fig. 7d shows the β weights for WT alone. The comparison of WT and KO is the essence of the manuscript— the authors should present these data for both genotypes. 3) In Fig. 7e, FMR1 KO weights for social context are similar to those of WT mice, and the differences in the other regressors are not significant. This is again not trivial given the title of the manuscript and the authors should directly address it.

REBUTTAL

We want to thank the three Reviewers for their positive feedback on our manuscript and their thoughtful comments and suggestions. We have done our best to address them individually, point-by-point, below. This necessitated additional analyses and changes to the manuscript and figures (including a new Main Figure 7 and six new Supplementary Figures 10-15). Notably, we implemented a linear encoding model to investigate the relationship between neural activity and avoidance/aversive behaviors in individual mice, we improved statistical comparisons, and we extended our decoding analysis to address potential differences in locomotion between contexts. There is no question that the manuscript is greatly improved after addressing the Reviewer's feedback.

Reviewer #1 (Remarks to the Author):

The study investigates the neural circuit mechanisms underlying the processing of social from non-social touch and how it's related to social interaction in Fmr1 KO mice, the model of Fragile X Syndrome, the most common form of autism. The authors have developed a novel social touch assay. While using this assay together with Neuropixels probes, they perform recordings in three brain areas: the vibrissal somatosensory cortex (vS1), the striatum (tStr) and basolateral amygdala (BLA). The authors describe neural correlates of social and non-social touch responses in voluntary and forced paradigms. Fmr1 KO mice demonstrate aversion to both social and non-social interactions. This observation also correlates with impaired encoding of social valence at the neural circuit level. WT mice demonstrated aversion to object touch but not to social touch, while Fmr1 KO mice showed aversion to both, indicating a potential neural basis for social avoidance in autism.

This manuscript is very extensive. The work is innovative and original, the methodology is rigorous with multiple controls and different characterizations.

We appreciate the Reviewer's praise of our manuscript.

There are some questions related to the manuscript:

1) It would be interesting to see if there is a statistical difference between the neural responses of different brain areas during social and object touch between WT and Fmr1 KO mice. It would be interesting to test if there a specific brain area responsible for this difference, too.

- We thank the Reviewer for this suggestion. We have now compared the magnitude of neuronal responses (z-score of the firing rate) averaged from 5 s stim period across genotypes under all touch conditions and used ANOVAs for statistical significance. This is shown in **Suppl. Fig. 11**. For WT mice under voluntary touch, we found that neurons in vS1 fired more to social touch (a similar trend for tSTR neurons), whereas under forced touch conditions, neurons in all three regions fired similarly to social and object touch. For *Fmr1* KO mice, we found that under voluntary touch conditions, neurons in vS1 fired also more to social touch, whereas under forced touch neurons in all three regions the opposite was true, that is they showed greater firing to object touch. Our interpretation is that forced object touch is a particularly salient stimulus for neurons across all regions in *Fmr1* KO mice.
- Overall, however, there were no statistically significant differences between WT and *Fmr1* KO mice in the activity of excited cells in any brain region, except for greater firing of vS1 neurons in *Fmr1* KO mice for forced object touch. This is now described in the Results section.
- Because forced object touch was the most aversive to *Fmr1* KO mice, we wondered whether greater firing of vS1 and tSTR neurons was associated with greater aversion at the behavior level in individual mice. We found generally weak correlations between the magnitude of touch-evoked neural activity (Z-score) and avoidance running or aversive facial expressions (at least when considering mean firing rates across individual mice). Only four such correlations were significant, and only under forced object touch: between the activity in vS1 and whisker protraction ($r=0.59$, $p=0.008$; Pearson's correlation analysis), as well as orbital area ($r=-0.45$, $p=0.053$), between activity in tSTR and orbital area ($r=0.644$, $p=0.003$), and between activity in BLA and orbital area ($r=0.46$, $p=0.048$), (see **Reviewer Figure 1** below). Because these correlations between activity and behavior only emerged when including data for both WT and *Fmr1* KO mice and were not seen for most types of presentations (social touch, voluntary object

touch), we did not include them in the paper. But please see linear encoding model to address points raised by Reviewer 3 (new Figure 7).

Reviewer Fig. 1: Correlation between firing rates (z-scored) and two aversive facial expressions. Each point is a different animal.

2) *The decoding accuracy of 60% doesn't seem to be very high. Does the accuracy of decoding increase with more cells or longer time? Is the high accuracy of decoding for the clusters excited by touch explained by the increased number of spikes? What about the clusters inhibited by spikes [sic]?*

- It is true that when looking over individual time bins of only 50 ms, the decoder accuracy is only 60-70% (**Fig. 3i**); but note that accuracy is clearly improved after touch (it goes as high as 75% in the forced touch condition). Our data is comparable to that of other papers using silicon probe recordings of neural activity (e.g., Yang and Masmanidis, *J Neurophysiol* 2020, PMID: 32727312). We suspect that decoding accuracy would increase if we used longer time bins, as is seen for studies using calcium imaging data. Even then, other published studies show similar decoding accuracy (e.g., Fig. 6 in Renard et al. *Nat Comm* 2022, PMID: 35780224; Fig. 5 in Buetfering et al. *Nat Neurosci* 2022, PMID: 36042310).
- When we trained classifiers on neural data averaged over 5 sec per each trial, decoding accuracy is much higher and reached >90% when 15 or more neurons are included in the analysis (**Fig. 3h**).
- The decoder performance was indeed highest for clusters that were excited by touch (e.g., Clusters 5, 10, 15) as shown in **Suppl. Fig. 7b-c**. However, clusters inhibited by touch (e.g., Clusters 2, 7, 13) also showed high decoding accuracy (well above chance). Moreover, neurons in clusters with the higher touch-evoked firing rates (e.g., Cl. 5, 10, 11, 14, 15) tended to show higher decoding accuracy. These points are now mentioned in the Results section.
- In general, we found clusters that showed significant differences in the modulation index by social vs. object touch (e.g., Clusters 2, 4, 5, and 10 for voluntary, 2, 4, 9, 11, 14 for forced; **Suppl. Fig. 6**) tended to show higher decoding accuracy (**Suppl. Fig. 7c-d**). To determine whether this was indeed a strong relationship, we plotted the magnitude of neural activity (z-score) against the decoder accuracy for each cluster in all WT and *Fmr1* KO mice and we found modest but significant correlations for most touch conditions ($r=0.31$ to 0.56 , $p=0.097$ to 0.001 ; **Reviewer Fig. 2**). This is to be expected though, since decoder accuracy was sensitive to firing rates.

Reviewer Fig. 2: Correlation between firing rates and decoder accuracy.

3) One of the important questions related to the manuscript is: are the same neurons encoding information about social touch also encoding information about object touch? Are these neuronal populations overlapping or not? Is the information encoded at the level of synapses or individual neurons?

- This is an important question, but unfortunately, our experiments did not allow us to look into that possibility because we did not record longitudinally the responses of neurons to the same stimulus across days. Note, also, that there was a 60 min break between social and object touch presentations in any given session.
- As shown in the pie-charts of **Fig. 4d**, most neurons in all three regions were not preferentially modulated by either social or object touch (gray region in pie charts). On the other hand, many neurons show a selectivity for either social touch or object touch at any given time, as calculated using ROC analysis. Of course, some neurons show a clear preference for one stimulus (e.g., they might fire a lot for social touch but not at all for object), while others show a more nuanced preference (they might fire more for social touch, but they also fire a little also to object). We then applied a strict cut-off (based on statistically significant differences - bootstrapping; see Methods) to identify object-preferring and social-preferring neurons. Such 'social' cells and 'object' cells did not overlap at the time when we recorded their activity.
- We suspect that some population drift exists in the encoding of social vs. object touch, meaning that a cell that fires preferentially to social touch on one session, might switch its preference a day later and fire more to object touch. Thus, we would favor a model in which the information (social vs. object) is encoded at the level of synapses rather than individual neurons. Depending on the inputs that a given neuron is receiving, its preference might change. We now expand on this point in the Discussion.

4) How is this information encoding impaired in *Fmr1* KO mice?

- One might imagine that, given the differences in how WT and *Fmr1* KO mice respond behaviorally to social vs. object touch, this might be reflected in differences in preference of neurons for social vs. object stimuli. We tested this explicitly using a two-way ANOVA test; however, we found no significant differences in the proportion of neurons showing a preference for social vs. object touch across brain regions between WT and *Fmr1* KO mice (compare Fig. 4e and Fig. 6e). Moreover, the similar aversion that *Fmr1* KO mice display to social and object presentations is not due to a higher % of neurons that show preference to social touch.

- Nevertheless, as we had already shown in **Fig. 6d**, there is a significant genotype difference in the mean ROC score, with vS1 neurons in WT mice showing a higher preference for social touch. Thus, *Fmr1* KO mice may not distinguish between social and non-social stimuli because neurons in vS1 show less preference for social stimuli than those in WT mice.

Minor comments:

1) In Supplemental Figure 10, is there a statistical difference between the modulation indices of WT and *Fmr1* KO mice in social compared to object touches? There seems to be a potential difference between the suppressed, but not excited units? Is this the case and if yes, what is the interpretation of this observation?

- In vS1, we found that suppressed cells are more strongly modulated by voluntary social touch than by object touch in WT mice (**Fig. 3f**). Although the magnitude of modulation was a bit smaller in *Fmr1* KO mice compared to WT controls, the difference was not significant (n.s. for voluntary touch, $p=0.052$ for forced touch). These results support our main interpretation that cortical neurons in *Fmr1* KO mice (whether excited or suppressed by touch) are not particularly recruited or engaged by social touch, i.e., that the unique salience that social touch represents for WT mice is not seen in the Fragile X model.
- The modulation index of neurons in individual clusters for WT and *Fmr1* KO mice can be found in Suppl. Figs. 6 and 12, respectively. Note that Cl. 2 (but not Cl. 1) in vS1 was clearly significantly more suppressed by voluntary social touch than by object touch in WT mice, but this was not the case in *Fmr1* KO mice. In contrast, suppressed neurons in cluster 7 of the tSTR, or cluster 13 of the BLA, did not show any preferential modulation by social vs. object touch in WT or *Fmr1* KO mice.
- Therefore, we conclude that it is generally excited neurons/clusters that are more strongly modulated by social touch in WT mice and that show the greatest genotype differences.

2) The description of the experimental paradigm for the voluntary touch vs forced touch is not very well explained.

- We now provide additional detail to explain the differences in the Methods section. We also refer the Reader to our previous paper that describes the paradigm in more detail (Chari et al., 2023; PMID: 37669860).

Reviewer #2 (Remarks to the Author):

Chari et al quantified aversive facial movements in wild-type mice and in mice with the *Fmr1* knockout, and evaluated the relationship between touch context and neural activity. **The results are novel and interesting, and advances our understanding of social interactions in mice especially wrt *Fmr1***, however I think more analysis is required to verify the decoder results and to disentangle facial movements and social valence.

We appreciate the Reviewer's praise of our manuscript.

1. The decoders were trained to discriminate two trial types (object vs social) that occurred in two separate blocks. Unfortunately the design of the experiment means that slow drift in neural activity could result in high decoding accuracy. Could you exclude neurons which had changes in average firing rate across the two blocks and still accurately discriminate? Also, given locomotion was different, could you discriminate trials with similar locomotor activity? This would check if the neurons/behaviors are coupled to locomotion.

- Yes, due to the nature of our experimental set-up and the need to swap what was being presented (object or mouse), the social and object presentations were done in different blocks, 60 min apart.
- The Reviewer is correct in pointing out that drift in neural activity could potentially explain differences in the decoding ability (although it would have been strange if all mice exhibited the same drift in baseline activity). To address these points and, as suggested by the Reviewer, we carried out additional analyses. We first performed the decoder analysis after excluding neurons

that showed large changes in baseline firing (no-touch condition, i.e., empty platform) between voluntary and forced touch sessions (New **Suppl. Fig. 7a**). We identified neurons that during the second presentation exhibited a change in baseline firing greater or smaller than 1.5 standard deviations of the mean in the first presentation session. Qualitatively, we found similar results for decoding accuracy of touch context (social vs. object), meaning a large increase in accuracy starting before touch, especially for forced touch conditions (compare **Suppl. Fig. 7a** and **Fig. 3i**; shown below for convenience). We now discuss this issue in the Results section.

Suppl. Fig. 7a: Decoder accuracy for touch context after excluding neurons with large changes in baseline firing between social and object presentations. (Fig. 3i, same but including all neurons, is shown for comparison)

- Next, we compared decoder performance between WT and *Fmr1* KO mice during trials with similar levels of locomotion, again as suggested by the Reviewer. Indeed, there were differences in how much time mice spent running or remained stationary during the stimulus presentations. Therefore, for this new analysis, we only included sessions in which mice were consistently running during all 40 presentations (i.e., any session that included stationary bouts was not considered). The selection left us with 4 WT and 5 KO mice for voluntary interactions, and 5 WT and 5 KO mice for forced interactions. This more stringent analysis confirmed our results and revealed significantly better decoding performance for neurons of WT mice in all three regions, especially vS1 and BLA. The only exception was for neurons in tSTR during voluntary touch, where the decoding accuracy was similar in WT and KO mice, but given the smaller sample size it is hard to reach a conclusion. This is now shown in **new Suppl. Fig. 14b** and mentioned in the Results section.
- Thus, we conclude that differences in neural activity between social and object touch are not a result of differences in locomotion levels across sessions, nor are they caused by slow drift in neural activity.

You could additionally build a decoder with locomotion+neural activity/whisking, and show that this decoder performs better than locomotion along [sic].

- This was a good suggestion. Because locomotion per se was not different between WT and *Fmr1* KO mice (as shown in Fig. 3b and Table 1 of our previous paper, Chari et al., J Neurosci, 2023), we suspected that adding locomotion to the neural data would not contribute greatly to improving decoding accuracy. As suggested by the Reviewer, we compared decoding accuracy for touch context (social vs. object) using classifiers that included or not locomotion and show the results in **Suppl. Fig. 7d** (shown below for convenience). We found that adding locomotion to neural data (pink line) did not significantly improve decoding accuracy compared to neural data alone (black line). Decoding with locomotion + shuffled neural data (blue line) was no better than decoding with shuffled neural data alone (gray line). We conclude that decoders based on neural data perform better than those based on locomotion behavior in decoding touch context.

Suppl. Fig. 7d: Decoder accuracy for touch context based on activity of 10 randomly selected cells with or without including locomotion (or shuffled locomotion) from each mouse.

Please also provide information about training the decoders, e.g. regularization coefficients (if different regularization coefficients were used for different sessions then a validation set is also necessary). Also, please try to define the decoder (e.g. object vs. social) in the figure panel titles (this may be less necessary when the captions are closer to the figures in the final paper).

- We now provide additional details in the Methods about default parameters used to train the classifiers.
- We now define the decoders in the figure panels, which will indeed help the Readers a lot.

2. Cluster analysis and sorting in Figure 3 should be performed in a cross-validated way: training trials used to find clusters/sorting, then average cluster activity plotted for test trials. This ensures that outlier trials do not drive the clustering. These test trials can then also be used to compute the modulation indices.

- Based on this feedback, we now clarify in the revised manuscript how the cluster analysis was performed in more detail (see Methods). We implemented a clustering approach based on firing averaged across all the trials, following what had been published previously in similar studies (Hocker et al., eLife 2021; Chirila et al., Cell 2022; Cox et al., Nat Comm 2016). We did not cluster based on a training dataset from a subset of trials and then cross-validate on test data. However, because 80 trials were averaged, we do not expect that rare outlier trials could drive the activity and influence clustering. Instead, to ensure that each neuron was being assigned to the correct cluster group, we performed and averaged multiple iterations of k-means (up to 10,000 times). Moreover, when we performed clustering of data from WT and *Fmr1* KO mice separately we found identical clusters for each brain region, so we believe it is unlikely that outlier trials would influence clustering. Finally, visual inspection of traces for each cluster (**Fig. 3e**) confirm that clustering identified distinct neural trajectories in response to touch.

3. I am wondering if the fact that the neural activity is not discriminative of social vs object context in the *Fmr1* mice is linked to the fact that the *Fmr1* mice makes more aversive movements to both compared to wild-type. This is difficult to disentangle but perhaps the authors could see if there are neurons correlated with aversive movements in the population and see if they are potentially more differentially modulated.

- This is a fair consideration. Indeed, we have previously reported that *Fmr1* KO mice manifest avoidance running and other movements in response to repetitive tactile stimuli (hypersensitivity). We attempted to perform an ROC analysis to identify neurons whose activity related to bouts of aversive behavior, but this did not provide a convincing result in part because the aversive responses occurred at the same time as touch, and also because some mice showed aversion in a disproportionately high proportion of touch trials, while others

demonstrated aversion in only a small proportion of trials. Nevertheless, we concluded that locomotion was not playing a major role in the genotype differences we observed because we did not see genotype differences in total locomotion. This was demonstrated in Fig. 3b and Table 1 of our previous paper, Chari et al., J Neurosci, 2023.

- See also new **Suppl. Fig. 14b** in response to this Reviewer's earlier comment (#1), where we found similar differences in decoder performance when we analyzed neural data during trials when WT and KO mice showed similar levels of locomotion.
- We also found that the modulation index of tSTR neurons was higher when mice exhibited aversive behaviors (**Fig. 3g**).

Reviewer #3 (Remarks to the Author):

In "A failure to discriminate social from non-social touch at the circuit level may underlie social avoidance in autism," Chari et al. aim to link differential behavioral responses to specific touch contexts (social/non-social) and touch choices (voluntary/forced) to differential neural responses evoked by the same stimuli in three brain regions: the somatosensory cortex (for sensory processing), tail of striatum (sensorimotor decision-making), and basolateral amygdala (emotional processing). The authors compare sensory-evoked behavioral and neural responses in wild-type mice to those observed in mouse models of autism (*Fmr1* KO) and describe reduced social/non-social discrimination in touch-evoked neural activity in *Fmr1* KO mice that could potentially underlie the increased aversion to social touch seen in ASD.

The question posed by the authors is compelling and aligns well with the existing literature on altered social categorization in mouse models of autism. The experimental design is straightforward and generally well-suited to addressing the research questions, despite the inherent challenges of using head-fixed mice to study social responses.

We appreciate the Reviewer's praise of our manuscript.

However, some inconsistencies in the presented data and certain analytical choices raise concerns about the authors' ability to fully support their main conclusion.

Major concerns:

1. Inconsistency between data and main claim: The paper's primary claim, as stated in the abstract and title, is that neurons in *Fmr1* KO mice "do not discriminate social valence." However, the data presented seem to contradict this assertion. For instance, in Supplementary Figure 10, both during forced touch and voluntary touch, neurons recorded in *Fmr1* KO mice are differentially modulated when presented with social versus object touch, indicating clear, significant discrimination between the stimuli. Specifically, both excited and suppressed units in vS1 of *Fmr1* KO mice show more modulation for social versus object voluntary touch, and the same is true for tSTR and BLA during forced touch. This result alone challenges the paper's main claim.

- We thank the Reviewer for this feedback. Accordingly, we have made changes to the text (including the Title and the Abstract) to provide a more balanced interpretation of our results. We have also performed additional analyses presented in new supplementary figures (see below). Overall, we believe that the important differences between WT and *Fmr1* KO mice that we highlight make our results significant and of broad interest to the neuroscience community. Below we provide more specific arguments in support of this conclusion.
- We agree with the Reviewer that vS1 neurons in *Fmr1* KO mice do show significantly greater modulation of their activity with social touch under voluntary conditions; but they do not at all under forced touch (**new Suppl. Figs. 10b and 10c**). This contrasts with results for WT mice, which show significantly higher modulation by social touch under both voluntary and forced conditions (**Fig. 3f**). Furthermore, even under voluntary touch conditions the difference in modulation by social vs, object touch (Δ modulation) is significantly smaller in *Fmr1* KO mice (see **Fig. 6a**).

- Results for neurons in the tSTR were interesting and surprising because of how differently they behave under voluntary vs. forced touch conditions. Note how tSTR neurons of WT mice switch their preference from being preferentially modulated by social touch under voluntary conditions to being preferentially modulated by object touch under forced conditions (**Fig. 3f**). In contrast, tSTR neurons of *Fmr1* KO mice were equally modulated by social and object touch under voluntary conditions (similar to what happens for vS1 neurons) but show a preference for object touch under forced conditions (**Suppl. Fig. 10b**). Thus, and in support of our claim, the complete swap in preference exhibited by tSTR neurons in WT mice is rather dramatic compared to *Fmr1* KO mice.
- In the BLA things are more difficult to interpret because neurons in both WT and KO mice seemed to be similarly modulated by social and object touch, with the exception that BLA neurons in *Fmr1* KO mice seemed to acquire a preference for object touch.
- We also show the modulation index for individual clusters from *Fmr1* KO mice in **new Suppl. Fig. 12**. Comparing directly with the same data from WT mice (**Suppl. Fig. 6**), it is striking to note that twice as many clusters showed significant differences in modulation index between social and object touch in WT mice (Cl. 2, 4, 5, 10 for voluntary interactions and Cl. 4, 9*, 11*, 14* for forced interactions) than in *Fmr1* KO mice (Cl. 3*, 5 for voluntary interactions and Cl. 10*, 11* for forced interactions; asterisks denote higher modulation index for object touch; all the others have more modulation for social). This is now mentioned in the Results.
- We also provide a direct side to side comparison of the difference in modulation index (social – object) between WT and *Fmr1* KO mice in **new Suppl. Fig. 13**.
- Overall, our data are consistent with neurons in vS1 and tSTR in *Fmr1* KO mice showing significantly reduced modulation by social touch compared WT mice. But we agree with the Reviewer that neurons in *Fmr1* KO mice do not demonstrate a complete loss of modulation. In that sense, we agree that it would be inaccurate to say that neurons in *Fmr1* KO mice “do not discriminate” social valence at all. *Fmr1* KO mice are simply *less able* to discriminate between social vs. object touch. To reflect a more faithful interpretation of our data, we have changed the title (and similar statements in the Abstract and Discussion) to: “a *reduced ability*” to discriminate.

Additionally, in Main Figure 6b, the authors present decoders for touch context (social vs. non-social) in wild-type versus *Fmr1* KO mice, with mean decoder performance quantified in Fig 6c. For two out of the three regions tested, decoder performance does not differ between the genotypes. In the other region (BLA), the *Fmr1* KO decoder performs worse than the wild-type, but still appears to be above chance levels, suggesting that some social discrimination is still apparent in this region for *Fmr1* KO mice. The authors should clarify how these results support their claim, as the data currently appear to contradict the paper's central thesis.

- It is true that the decoders using neural data from *Fmr1* KO mice do perform better than chance. But decoder accuracy is still higher for WT mice, which supports the main conclusions of the paper. When using statistics for different mice, the differences did not reach significance for vS1 or tSTR, but the differences are more apparent when looking at cell averages, which is particularly striking in **new Suppl. Fig. 14b** (shown below for convenience), when looking at only mice that showed similar amounts of locomotion in both presentations.

Suppl. Fig. 14b: Decoder accuracy for touch context based on activity of 10 randomly selected cells from WT and *Fmr1* KO mice that were consistently running across both presentation types.

- The Reviewer is correct in pointing out that the quantification of decoder performance did not identify significant genotype differences for vS1 and tSTR neurons when using strict statistics with $N =$ number of mice as the sample size. We further emphasize this finding in the Results section. However, the accuracy of the BLA decoder, which has the highest decoding accuracy in WT, was significantly lower in *Fmr1* KO mice.
- A confound might be that some genotypes might have stereotyped locomotor responses to certain stimulus presentations. We addressed this point comparing decoder performance between WT and *Fmr1* KO mice during trials when they exhibited similar levels of locomotion, as suggested by Reviewer 2. We once again found significantly better decoding performance for neurons of WT mice in all three regions, especially vS1 and BLA. This is now shown in **new Suppl. Fig. 14b** (see above).
- To summarize, in the revised manuscript we have provided additional data to support our main conclusions regarding genotype differences, while also providing a more nuanced interpretation of these results, which we hope addresses the Reviewer concerns.

2. Lack of direct correlation between behavioral and neural data: Another key claim of the paper is that differential neuronal responses underlie differential behavioral responses to touch stimuli. However, the authors do not directly correlate behavioral and neuronal data, except for a single panel (Fig. 3g). Instead, they analyze the behavioral response to a stimulus and the neuronal response to the same stimulus independently, leaving readers to infer a brain-behavior correlation that is not explicitly derived from the data. Since the authors recorded behavioral and neuronal responses simultaneously, if they wish to claim that specific neural responses underlie specific behavioral responses, they should present a direct correlation.

For example, can neuronal responses predict behavioral aversion levels on a trial-by-trial basis, rather than merely predicting stimulus context or choice? If aversion parameters are similar (regardless of stimulus context or choice), are the neural responses similar? Or, per animal—compared to some average or baseline modulation factor—when an animal's response to social touch is more aversive, is the neural response more similar to that evoked by an object? This approach would not only strengthen

the paper's claims regarding genotype differences but also provide a more direct correlation between neuronal activity and aversion levels within each genotype. Overall, a manuscript that claims a relationship between behavior and neural activity should demonstrate this link explicitly.

- We fully agree that establishing a more direct relationship, or correlation, between neural activity and behavior would strengthen the paper. When we calculated Pearson correlations between neural firing rates (or modulation index) and behavior features, we only found significant correlations between neural activity (Z-score) in vS1 and whisker protraction, and between activity in tSTR or BLA and orbital tightening (see our response to Reviewer 1 and **Reviewer Fig. 1** above; using N = number of mice).
- We therefore implemented a linear encoding model to predict trial-averaged neural activity from behavioral variables (regressors), including the aversive behavioral responses that we had quantified. Although this had been our plan for a follow-up study, to address the Reviewer's comment, we provide initial results with this model. The model identified neurons whose activity nicely matched either stimulus context (social vs. object), running, or AFEs (see examples in new **Fig. 7b**). We found significant positive correlations between the β weights of single neurons for social context and their β weights for orbital tightening in WT mice (**Fig. 7c**). In contrast, neurons from *Fmr1* KO mice tended to show weaker correlations between those β weights, or the inverse correlation. When we compared β weights for different regressors across brain regions in WT mice, we found higher values for context, orbital tightening, and whisker movement than for pupil size in vS1 (**Fig. 7d**). In the tSTR and the BLA, β weights were also larger for some AFEs, but there was a lot more variability. Importantly, in vS1, β weights for orbital tightening and whisker motion were, on average, significantly higher in WT mice than in *Fmr1* KO mice (**Fig. 7e**).
- We intend to extend these preliminary results in a subsequent paper and to expand the current model to account for dynamics that occur within individual trials. We scrutinize some of the caveats of the current analysis on the Discussion section.
- From this encoding model analysis and by comparing the average weights between animal groups (**Fig. 7e**), we conclude that neural activity in vS1 encodes not only touch context, but also certain AFEs in WT mice. In contrast, neural activity appears to be less predictive of behavioral response in *Fmr1* KO mice.

3. Analytical choices: The authors make several analytical choices that could be reconsidered:

Incomplete data presentation: The data presented in the figures do not always fully correspond to the experimental conditions defined in the paper (social/non-social, voluntary/forced, wild-type/KO), making it difficult to evaluate the consistency of the results. For example, Figure 1f, h panels are presented only for voluntary touch; Figure 1l (left) only for social touch; Figure 2 only for wild-type mice, not *Fmr1* KO; Figure 3e only for wild-type, not KO; Figure 3g only for excited neurons; Figure 3h, i only for voluntary, not forced touch; Figure 5d only for voluntary; and Figure 6e only for forced touch, with no direct comparison to wild-type. Although the authors can choose which data to present in the main figures and which in the supplementary materials, the complete analysis of the datasets should be presented somewhere in the manuscript (again, in the context of the chosen experimental groups) if the results are to be considered supportive of the conclusion.

- We appreciate this feedback and present a revised version of the paper to include all the data the Reviewer requested:
 - For Fig. 1f-h, the data for decoder of touch context (social vs. object) using all DLC labels are now shown for forced touch in Suppl. Fig. 1c.
 - For Fig. 1l (left), the data for decoder of touch choice (voluntary vs. forced) for object touch is now shown for forced touch in Suppl. Fig. 1d.
 - For Fig. 2, we never performed TRAP experiments in *Fmr1* KO mice.
 - For Fig. 3e-f, we now show the same data for *Fmr1* KO mice in new Suppl. Fig. 10a-b.
 - For Fig. 3g, we did not find significant differences in the M.I. for vS1 suppressed cells, as shown in **Reviewer Fig. 3** below.

Reviewer Figure 3: Modulation index for vS1 suppressed cells (voluntary touch); relates to Fig. 3g

- For Fig. 3h, we had already shown the decoder accuracy as a function of the number of neurons used for both voluntary and forced touch in Fig. 3h.
- For Fig. 3i, we now show the decoded accuracy as a function of time for both voluntary and forced touch in Fig. 3i.
- For Fig. 5d-g, we now show the equivalent data for decoder accuracy of touch context and touch choice for forced touch interactions in new Suppl. Fig. 9b-g.
- For Fig. 6e, we now show the % of social-preferring vs. object-preferring cells for *Fmr1* KO mice during both voluntary and forced interactions.

Simplified neural data analysis: The authors' decision to pool together different clusters of units with distinct response patterns may obscure specific neuronal specializations that are relevant to the experimental questions. For example, in Figure 3e, cluster 9 and 10 are pooled together even though their response patterns differ substantially in terms of both dynamics and social preference. By averaging/pooling the clusters, the authors present results that are both not specific enough and could wash out neuronal specializations that are actually very relevant for the experimental questions. For instance, in cluster 10, the difference between social and non-social is much larger than in all other clusters—a finding that can be either muted for this cluster by pooling with other clusters, or increased overall specificity when averaged with clusters not showing such preference.

- This is a fair point, and we agree that pooling data could obscure some interesting results for specific clusters. Based on this feedback we now present data for individual clusters in **new Suppl. Figs. 12 and 13**.
- In a way, this comment gets to a very important of brain function: is it about the activity of individual neurons, or small groups of neurons that fire in a similar way, or perhaps what matters is the overall global input of a particular brain region (lumper vs. splitter debate). One downside of considering only individual clusters is that they reflect unique activity in a snapshot in time, and we do not know that they represent true functional ensembles that encode specific behaviors. The activity of neurons is fluid and a neuron belonging to one cluster now might belong to a different cluster later.
- Ironically, some of our colleagues at UCLA had advised us to dispense with clusters and just pool all the neurons in each brain region. The simplest way was to focus on excited/suppressed neurons, which is what we show in **Fig. 3c**. But we felt this was not sufficient because it ignored richer temporal dynamics observed in the raster of all neurons (**Fig. 3b**), so we tried the clustering approach. Both approaches are valid.
- We agree with the Reviewer that certain clusters are uniquely modulated by social touch, like Cl. 10 in the tSTR; but note that it had a small number of neurons. One pitfall of focusing too much on clusters is that the sample size of neurons is smaller, which weakens statistical power. In the end, we did not find anything very compelling when looking at individual clusters. In a way, we hope that the Reader will be convinced by the fact that region-wide differences can emerge even when data from different clusters is pooled. For example, the opposite modulation

by social or object touch of tSTR excited cells depending on touch choice (voluntary or forced; **Fig. 3f**) may be driven primarily by neurons in Cl. 10 and Cl. 11, respectively, but it is still present when pooling all excited neurons together.

In Supplementary Figure 6, when the authors explore the social discrimination of neurons recorded in wild-type (wt) mice, only very few clusters of neurons (how many neurons overall is unclear) show differential responses to social versus nonsocial touch. These are pooled together with clusters that do not show significant preference to social valence, which implies that even in wt mice, the discriminability is not that prominent. This pooling may obscure important distinctions and undermine the specificity of the findings. The authors have many other options—such as using only clusters with the most differential modulation between social and non-social stimuli—that could provide a clearer understanding of the data. Can the authors explain their analytical choice?

- We now show the modulation index, and the delta-modulation index (social-object) for all clusters in WT mice and in *Fmr1* KO mice (see **Suppl. Fig. 6** and **new Suppl. Figs. 12 and 13**). We also indicate the number of neurons in each cluster in the relevant figures, as requested.
- We respectfully disagree with the Reviewer's conclusion that "*even in WT mice, discriminability is not that prominent*" because we find statistically significant differences in the modulation of excited neurons by social vs. object touch (**Fig. 3f**). The fact that these differences are robust even after pooling different clusters underlines the robustness of these findings.
- In the end we chose to pool excited neurons from different clusters because the trends for these clusters were similar. For example, Readers of the paper will see how neurons in clusters 3-5 in vS1 behave similarly under voluntary touch (they are more modulated by social touch), or how those in clusters 9-11 in tSTR also behave similarly under forced touch (they are more modulated by object touch). Some of these differences were not always significant for individual clusters, but this did not affect the results of pooled data.
- The biological significance of the different behaviors of neurons in individual clusters is presently unclear. As stated above, the activity of neurons in each cluster represents only a snapshot in time. Due to population drift, cells exhibiting a particular behavior at one time (which led us to classify them as belonging to a given cluster) may show a different behavior at a later time (and would have been classified into a different cluster).

Lack of data on stability, specificity and dynamics over time: The authors do not provide enough data to understand the stability or specificity of neural responses. For example, can neurons that are more modulated by the object voluntary condition (vs. object) be more modulated by the social forced condition (vs. object)? How stable is the neuronal modulation score? What about preference measurements over trials? Are these measures different across regions or genotypes? How do they correlate with the consistency and specificity of behavioral responses? What are the population sparseness and lifetime sparseness values? How do the dynamics of the different response types differ, and what is their correlation with dynamics of behavioral responses?

- This is a very interesting question (also raised by Rev. 1) and we agree that it would be worth pursuing. We did not initially compare the same units across voluntary and forced conditions, as these were performed on different days, because we could not be confident that we could track the same units on two different days. We intend to pursue this question in future studies when we track the same units across multiple days.

Insufficient Data on KO Mice: The authors also show substantially less data for the neural recordings conducted in KO mice. All the results presented for wild-type neuronal responses should also be presented for units recorded from KO mice.

- We now show additional supplementary figures for KO mice to match WT results, as discussed above in response to the Reviewer's earlier comment.

4. Framing of the paper experimental design: Some aspects of the paper's framing seem somewhat arbitrary. For instance, "social" and "non-social" are broad categories and should not be generalized based on a singular stimulus. The authors test only one social and one non-social stimulus, even though previous work from the same group has shown differential responses to different types of social stimuli.

It is possible that the observed results are related to other tactile properties of the stimuli rather than their social/non-social nature (e.g., the tube is smooth, the mouse is not; the tube is still, the mouse is moving). In that sense, "social vs. non-social" is really "mouse vs. tube."

Additionally, the terms "voluntary" versus "forced" touch might need re-examination, as the distinction is not as clear-cut as implied. The concern is that "voluntary" versus "forced" is only "close but not touching" versus "touching your nose." These are not solely semantic distinctions but have a direct effect on the claim of the paper. The authors might consider adding more types of social and non-social stimuli or rethinking their definitions to solidify their arguments.

- We understand the Reviewer's point of view. Naturally, we considered several other possibilities and reflected at length on which terms best described the types of interactions mice were subjected to in this assay. The terminology we chose is the same as the one we used for our previous paper (Chari et al., 2023) so it would not make sense to change it at this point.
- Note that the 'social' vs. 'object' distinction matches terminology that is frequently used in the literature for related studies using similar assays (Everts and Koolhaas, Brain Research 1997, PMID: 9237511; Bobrov et al., Curr Biol 2014, PMID: 24361064; Ebbesen et al., Nat. Neurosci, 2017, PMID: 27798633; Jennings et al., Nature 2019, PMID: 30651638; Jeon et al., J Neurosci 2023, PMID: 36717231).
- For the non-social stimulus, we selected "object" as a term that was perhaps not as accurate as "a 50 cc polypropylene Falcon conical centrifuge tube", but indeed appropriately described a generic category for the animal that was not "social". And it matches what is typically used in the literature.
- We agree that there are other differences between the two types of presentations (texture, movement, smell). In our previous paper, we did use different types of inanimate objects as the non-social stimulus (the same plastic tube we used here and a plush toy mouse) and found no differences in how WT or *Fmr1* KO mice responded to each "object", which is why we did not add additional objects here. Certainly, in future studies we aim to compare differences in behavioral responses (and neural activity) for different social scenarios, like using familiar vs. stranger mice, or responses of a female dam to one of its pups. We now mention this in the Discussion.
- As far as the terms *voluntary* and *forced*, again, we chose to be consistent with our previous paper. Although these two scenarios could be described in several different ways, referring them here as 'close but not touching' vs. 'touching your nose' is also not ideal (but yes, we understand the Reviewer) and, more importantly, it would not reflect our ultimate intent. Indeed, we designed our paradigm with the goal of testing two types of social interactions, one in which the animal has a choice to engage (because the presented stimulus is at a distance that requires the test animal to protract its whiskers), and the other when the animal has no choice but to interact, whether it likes it or not, because the visitor is in direct contact with its snout.

Finally, a minor but noteworthy point is the use of the terms "personal space" and "aversion." It appears that all the stimuli presented in this study are aversive, at least based on the behavioral responses. Is this truly the case? The authors might consider including a pleasant or attractive stimulus to provide a broader perspective. Additionally, while the notion of "personal space" is frequently mentioned, it is neither empirically explored nor quantitatively defined in the study. It would be helpful to clarify the relevance of this concept to the paper's findings, or reconsider its inclusion if it does not add substantial value.

- We too wonder how mice would respond to a stimulus that is pleasant or attractive to them. But no, not all stimuli were aversive and, in fact, we consider that voluntary social touch is perhaps attractive to WT mice, since they do not manifest any aversive behaviors to it. Please note that the levels of avoidance running, aversive whisker protraction, or orbital tightening we observed for WT mice under voluntary social interactions represent the baseline of the measurement (Fig. 1d) and are not an indication of mild aversion. As mentioned above, we hope to further explore presentations of more 'pleasant' stimuli in future studies, for example by presenting to a female mouse one of its pups, or by presenting a plastic tube containing a sweet treat (like peanut butter chips).
- There is great interest in the topic of personal space in the context of autism. Our experimental design allowed us to examine this issue. The behavioral differences we observe in *Fmr1* KO mice for social touch when choice is involved (voluntary vs. forced) are striking (**Fig. 5c**). While

perhaps not a definitive exploration of this aspect of social interactions, we consider it a valuable contribution of our paper that will hopefully inspire others to pursue it in their own studies of social behaviors in mice. We now address this in the Discussion.

Other Concerns:

1. Decoding comparisons: The authors present various decoders, but many lack comparison with shuffled data, making it difficult to assess whether the decoding performance is above chance levels or differs between wild-type and *Fmr1* KO mice. For example, in Figures 5d, 5g, and 6b, there is no comparison with shuffled data. The authors should ensure that decoding performance is always compared to chance and explore whether it exceeds it.

- We have now added shuffled data to Main Figs. 5d, 5h, 6b, and Suppl. Figs. 7b, 7e, 9b. The decoder accuracy for shuffled data always hovers at ~50% as expected.

2. Clustering analysis: The clustering analysis of the neural data raises some questions. It is only presented for wild-type and not *Fmr1* KO mice, and the justification for the number of clusters chosen is not clearly explained. The authors should provide more clarity on how clusters were identified and whether the same criteria apply to *Fmr1* KO mice. Additionally, in Supplementary Figure 7, the authors discuss the "top-performing clusters" for decoding social context, but these clusters seem inconsistent with the data presented in the main figure 3. For example, in Supplementary Figure 7e, the authors claim that cluster 6 is most informative for forced touch, yet in Figure 3 and the main text, cluster 6 is defined as "unmodulated" by the stimulus social context. Similarly, cluster 14, rather than 15, seems to be more modulated by social context, although transiently. The authors should explain this inconsistency.

- To determine the number of clusters for each brain region, we did the clustering after combining the neural data from WT and *Fmr1* KO mice. However, even when we did it separately for each genotype we found the same clusters, suggesting that the chosen number robustly reflects the dimensionality of the dataset. We have expanded on the description of the clustering approach in the Methods section.
- Despite the slight differences in performance of individual clusters seen in Fig. 7d, note that they all perform extremely well (reaching >80% accuracy).

3. Statistical analysis: The statistical analysis and conclusions drawn from it should be revised for consistency.

- Although the methods suggest that the authors apply reasonable statistical approaches to the data, throughout the paper, the authors do not treat the value of $\alpha < 0.05$ as a strict threshold. As a result, p-values larger than 0.05 are sometimes considered in the paper (implicitly or explicitly) as significant effects supporting the authors' claim. The main issue here is that in such cases, the authors have the discretion to consider the effect relevant or not—a situation that the setting of a statistical threshold aims to avoid. The authors should determine their threshold and adhere to it throughout the paper when discussing their results.
 - We appreciate the Reviewer pointing this out. We were merely trying to emphasize interesting trends in the data that matched other significant differences and let the Readers reach their own conclusion. In the revised manuscript, we have removed all the p values that approached significance throughout, with only 4 exceptions where we believe interesting trends are indeed important to point out: in Fig. 1d ($p=0.051$), Fig. 3d ($p=0.055$), Fig. 5c ($p=0.054$), and Suppl. Fig. 10c ($p=0.052$). This is mentioned in the Methods.
- The results of many of the statistical analyses are not fully presented; only the post-hoc t-tests are shown. It is crucial to determine whether factors such as social context and genotype show a main effect or interaction effect before considering the post-hoc comparisons. Some of the post-hoc comparison results are also unreported. The authors should clearly present all statistical analysis results to ensure transparency and comprehensiveness.

- We had only presented p values for post-hoc comparisons when there was a statistically significant main effect (of genotype or touch type). We now include a new Table with all the relevant statistics for each figure.
- Additionally, any conclusions regarding differences between wild-type and *Fmr1* KO mice should be drawn from direct comparisons between the genotypes, rather than from independent analyses of each. Consistency in this approach across the manuscript would strengthen the validity of the findings.
 - We have made an effort to include the same data for mice of both genotypes. There are only three panels in the main figures where we consider *Fmr1* KO mice alone. Two of these relate to aspects of behavior (Fig. 4a-b) for which we had already shown statistically significant genotype differences in separate experiments published in our previous paper (Chari et al., J Neurosci 2023). The other was for the proportions of neurons that showed preference for social vs. object touch and we now show the *Fmr1* KO data (**Fig. 6e**) in the same way we show WT data (**Fig. 4e**). We believe we present all the data in a transparent way.

4. Importance of platform movement: The authors dedicate some analysis to "before touch" and "after touch" decodability. However, the importance of this analysis is not entirely clear. Isn't it expected that decodability of touch would increase once touch is initiated? If the key finding is that this phenomenon differs in *Fmr1* KO mice, the authors should emphasize this specific comparison and discuss its implications in greater detail.

- Yes, we agree that one would expect that decoding accuracy would increase upon contact, and that is exactly what is seen in WT mice (**Fig. 1h, j-m**). In contrast, for *Fmr1* KO mice the decoder performance did not change after voluntary touch (**Fig. 5d-e**). We can only speculate as to why, but one possibility is that touch-evoked neural activity is unable to discriminate social from non-social stimuli in *Fmr1* KO mice, so they show similar levels of aversion to both. Hence, the mere fact that something is approaching is aversive to them. We now mention this in the Discussion, as suggested by the Reviewer.

REBUTTAL

We want to thank the all the Reviewers for their positive feedback on our revised manuscript. Reviewer 3 had three minor comments that we address below.

1) *In Fig. 7b, the distribution of β weights clearly shows that the activity of the selected units reflects contributions from several regressors. The authors should reconsider the use of the term “specific” here.*

- We have removed the term “specific” from the figure panel, and we now say “modulated” (for example, “Context-modulated cell”).

2) *Fig. 7d shows the β weights for WT alone. The comparison of WT and KO is the essence of the manuscript— the authors should present these data for both genotypes.*

- We have included the equivalent data for Beta weights from *Fmr1* KO mice for easier comparison with the WT data (and eliminated Fig. 7e)

3) *In Fig. 7e, FMR1 KO weights for social context are similar to those of WT mice, and the differences in the other regressors are not significant. This is again not trivial given the title of the manuscript and the authors should directly address it.*

- We used an ANOVA to compare beta weights across all regressors for WT and *Fmr1* KO mice in revised Fig. 7d. We found a significant interaction effect (regressors x genotype) in vS1 ($p=0.006$; ANOVA), but not in tSTR or BLA. We now explicitly state that, although beta values for the two main AFE regressors (orbital area and whisker DLC) appeared to be higher in WT mice this did not reach statistical significance.
- This does not change the conclusions of our paper. In fact, our title had already been changed in response to the Reviewer’s previous review to indicate that a reduced ability of circuits to discriminate social from non-social touch may underlie social avoidance.